

# Morphological transformation of soot: investigation of microphysical processes during the condensation of sulfuric acid and limonene ozonolysis product vapors

Xiangyu Pei[1], Mattias Hallquist[1], Axel C. Eriksson[2,3], Joakim H. Pagels[3], Neil M. Donahue[4], Thomas Mentel[5], Birgitta Svenningsson[2], William Brune[6], Ravi Kant Pathak[1]

[1]Department of Chemistry and Molecular Biology, University of Gothenburg, Gothenburg, 41296, Sweden
[2]Division of nuclear physics, Department of Physics, Lund University, Lund, 22100, Sweden
[3]Ergonomics and Aerosol Technology, Lund University, Lund, 22100, Sweden
[4]Center for Atmospheric Particle Studies, Carnegie Mellon University, Pittsburgh, PA, 15213, USA
[5]Forschungszentrum Jülich GmbH, Jülich, 52428, Germany
[6]Department of Meteorology and Atmospheric Science, Pennsylvania State University, University Park, PA, 16802, USA

*Correspondence to*: Ravi Kant Pathak (ravikant@chem.gu.se)

**Highlights:**

1. Morphological transformation occurs via two key complementary and sequential processes, i.e., void filling in the particle and its diameter growth

2. A framework was developed for quantifying the state of morphological transformation of soot, i.e., the utilization of material for filling and growth during the condensation processes

3. A method was developed for (i) quantifying the fraction of internal voids and open voids in the soot agglomerate, (ii) deriving the volume equivalent diameter (inclusive of internal voids), and (iii) estimating the *in-situ* dynamic shape factor

**Abstract.**

The morphological transformation of soot particles via condensation of low-volatility materials constitutes a dominant atmospheric process with serious implications for the optical and hygroscopic properties, and atmospheric lifetime of the soot. We consider the morphological transformation of soot agglomerates under the influence of condensation of vapors of sulfuric acid, and/or limonene ozonolysis products. This influence was systematically investigated using a Differential Mobility Analyzer-Aerosol Particle Mass Analyzer (DMA-APM) and the Tandem DMA techniques integrated with a laminar flow-tube system. We discovered that the morphology transformation of soot results (in general) from a two-step process, i.e., (i) filling of void space within the agglomerate; (ii) growth of the particle diameter. Initially, the transformation was dominated by the filling process followed by growth, which led to the accumulation of sufficient material that exerted surface forces, which eventually facilitating further filling. The filling of void space was constrained by the initial morphology of the fresh soot as well as the nature and the amount of condensed material. This process continued in several sequential steps until all void space within the soot agglomerate was filled. And then "Growth" of a spherical particle continued as long as vapors condensed on it. We developed a framework for quantifying the microphysical transformation of



soot upon the condensation of various materials. This framework used experimental data and the hypothesis of 'ideal sphere growth' and void filling to quantify the distribution of condensed materials in the complementary filling and growth processes. Using this framework, we quantified the percentage of material consumed by these processes at each step of the transformation. We also used the framework to estimate the fraction of internal voids and open voids. This information was

then used to derive the volume equivalent diameter of the soot agglomerate containing internal voids and to calculate the *in-situ* dynamic shape factor. The dynamic shape factor estimated based on the traditional assumption (of no internal voids) differed significantly from the value obtained in this study. Internal voids are accounted for in the experimentally derived *in-situ* dynamic shape factor determined in the present study. In fact, most of the fresh soot particles considered in this study were largely spherical (dynamic shape factor: ~1.1). The effective density was strongly correlated with the morphological

transformation responses to the condensed material on the soot particle and the resultant effective density was determined by the (i) nature of the condensed material; (ii) morphology and size of the fresh soot. This work constitutes the first study that quantitatively tracks *in-situ* microphysical changes in soot morphology, providing details of both fresh and coated soot particles at each step of the transformation. This framework can be applied to model development with significant implications for quantifying the morphological transformation (from the viewpoint of hygroscopic and optical properties) of

soot in the atmosphere.

## 1 Introduction

Soot containing black carbon (BC) produced from incomplete combustion of fossil-fuel or biomass is ubiquitous in the atmosphere, and represents a major constituent of atmospheric aerosols (Bond et al., 2013). Through direct radiative forcing, soot influences the climate significantly as it efficiently absorbs light in the visible and infrared spectral ranges.

Once emitted into the atmosphere, soot particles undergo morphological transformations due to several ageing processes. Soot ageing includes adsorption and condensation of semi-volatile vapors, coagulation with pre-existing aerosol particles, heterogeneous reactions with atmospheric gaseous species, and in-cloud processing (Khalizov et al., 2009). The physical and chemical properties of soot particles change considerably during these ageing processes. For example, exposure to sulfuric

acid changes the morphology, enhances the absorption and scattering, and increases the hygroscopicity of fresh soot particles (Zhang and Zhang, 2005; Zhang et al., 2008). The cloud forming properties of soot can be altered, and climate forcing can be influenced (albeit indirectly), by coating the soot with soluble material. For example, atmospheric ageing of coated soot enhances the hygroscopicity of the coatings and, hence, aged soot particles can act as efficient cloud condensation nuclei (CCN) or ice nuclei (IN) (Ammann et al., 1998; Henning et al., 2012). Transformation from fresh hydrophobic soot to

processed hygroscopic soot can also have adverse health effects as, compared with hydrophobic fresh soot, aged hygroscopic soot is more efficiently deposited in the lungs (Löndahl et al., 2008).





Soot particles are emitted as complex agglomerates. Their morphology and life-span in the atmosphere are directly correlated with the atmospheric processing (e.g., the type and the amount of material condensed on the agglomerate). The mass mobility exponent (*Dfm*) is an indirect measure of the morphology of irregularly shaped agglomerated particles (DeCarlo et al., 2004). The DMA-APM technique characterizes the properties of fresh or coated soot agglomerates by
performing simultaneous measurements with a differential mobility analyzer (DMA) and an aerosol particle mass analyzer (APM) (McMurry et al., 2002). With this technique, the mobility diameter and mass of the same agglomerate particles are accurately measured, and the mass mobility exponent is determined through fitting of the mass-mobility relationship. The mass mobility exponent, *Dfm*, is defined as:

$$Dfm = \frac{\ln(m_p) - \ln(k)}{\ln(D_p)},$$ (1)

where $D_p$, $m_p$, and $k$ are the particle mobility diameter, particle mass, and proportionality constant, respectively.

The effective density of a particle, $\rho_{eff}$, can also be determined from simultaneous measurements of the mass and mobility diameter, and is given as follows:

$$\rho_{eff} = \frac{6m_p}{\pi D_p^3},$$ (2)

where $m_p$ is the particle mass and $D_p$ is the mobility diameter.

A *Dfm* of 3.0 is obtained for spherical particles, whereas values of 2.2–2.4 are typically obtained for fresh diesel soot (Park et al., 2002). This range of values is indicative of an open-structure aggregate. Schneider et al. (2006) found that particles resulting from biomass burning have variable *Dfm* values: for dry beech fuel *Dfm* = 2.1, whereas for humid oak fuel *Dfm*
was close to 3.0. Fresh soot from various sources, such as a diesel engine, diffusion-type burner, and premixed-type burner, have effective densities of 0.2–1.2 g cm$^{-3}$, which typically decrease with increasing mobility diameter (Ghazi et al., 2013; Rissler et al., 2013).

Previous studies (Zhang et al., 2008; Weingartner et al., 1995) have found that the mobility diameter of fresh hydrophobic
soot below water saturation (relative humidity < 100%) undergoes a relatively small amount of hygroscopic growth. This indicates that, at sub-saturation, water vapor has little influence on the soot morphology. However, soot particles will become hydrophilic if water-soluble compounds, such as sulfuric acid or organics condense on the surface, and then collapse into a spherical shape when the relative humidity (RH) is increased (Zhang et al., 2008; Pagels et al., 2009). The effective density of coated and annealed soot increases, consistent with restructuring of the soot cores after they acquire sufficient
fractions of coating material (Cross et al., 2010; Qiu et al., 2012; Saathoff et al., 2003; Pagels et al., 2009). The extent of restructuring depends on the production conditions of the soot, as well as on the mass fraction and type of the coating material. Carbon soot aggregates produced from spark discharge collapse to relatively compact shapes at sub-saturation, whereas diesel soot undergoes limited restructuring (Weingartner et al., 1997). Different dicarboxylic acids have different



effects on soot restructuring. Exposure to glutaric acid ($C_5$ low $RH_{deliquescence}$) leads to significant collapse of the soot cores, whereas condensation of succinic acid (C4, high $RH_{deliquescence}$) has no influence on the restructuring (Xue et al., 2009). Secondary organic aerosol (SOA) produced from OH-initiated oxidation of both toluene and isoprene results in an increase in the effective density and a significant decrease in the dynamic shape factor of the cores (Qiu et al., 2012; Khalizov et al.,

2013). However, the effect of limonene SOA on soot ageing has yet to be studied, despite the fact that limonene comprises up to 10% of the total global monoterpene emissions (Geron et al., 2000; Guenther et al., 2012) and has very high potential for SOA formation. Limonene is efficiently oxidized to SOA with little or no carbon loss, generating water-soluble products (Pathak et al., 2012) due to two different unsaturated bonds in the molecule: an endocyclic tri-substituted double bond and an exocyclic terminal double bond. In this work, we determine the effect of limonene ozonolysis products on soot morphology.

Sulfuric acid – formed during the combustion processes, condenses on soot and provides active acidic surfaces for the condensation of semi-volatile SOA. The acid-catalyzed SOA reactions can enhance SOA formation significantly, as shown in laboratory experiments (Jang et al. 2002). However, soot – SOA interactions mediated by surface acidity have barely been investigated. For e.g., the effect of an SOA-added acidified soot surface on the morphology, density, and, hence, the lifetime

of soot in the atmosphere as well as the radiative properties remains unexplored.

Condensation of such low-volatility materials on soot particles leads to morphological transformation of these particles with serious implications for the corresponding optical and hygroscopic properties, and atmospheric lifetime. Nevertheless, many aspects of this transformation (including the mechanism of void-space filling within fractal soot and the restructuring and *in-*

*situ* growth upon condensation of different materials) remain unclear. The correlation (i.e., mutual dependence and/or sequential dependence) or lack thereof between these processes must also be clarified. To date, these microphysical aerosol processes are neither described in current models nor fundamentally studied and quantified. Therefore, major knowledge gaps remain, thereby hindering the development of robust modeling tools for improved climate predictions.

To address these issues, a study of chemical and physical processes encompassing the interaction of SOA with soot on the acidic surfaces is performed using a flow-tube reactor at the University of Gothenburg, Sweden. In this study, soot processing is investigated using controlled condensation of sulfuric acid and SOA onto a laboratory-generated propane flame soot. We perform direct measurements of the change in mobility diameter and mass upon processing, using the TDMA and DMA-APM techniques. The mass-mobility relationship is then used to calculate the mass mobility exponent, effective

density, and dynamic shape factor of soot subjected to processing of atmospheric relevance. Subsequently, the results are compared with those of previous studies. A framework is developed for quantifying the morphological state and subsequent transformation of soot, i.e., the utilization of material for filling and growth upon the condensation of material. Using this framework, a method is derived for (i) quantifying the fraction of internal voids and open voids in the soot agglomerate, and (ii) determining the volume equivalent diameter inclusive of internal voids.



## 2 Experimental methods

Soot particles emitted from a premixed-diffusion propane flame soot generator with and without coatings were studied. The coating processes were systematically performed in a temperature and RH-controlled laminar flow tube as shown in Fig. 1. The key instruments used included an APM, two DMAs, and two condensation particle counters (CPCs). A schematic of the experimental setup is shown in Fig. 1.

### 2.1 Soot generation

Laboratory-scale soot-SOA interaction process experiments were performed using a versatile propane premixed-diffusion flame soot generator built, in-house, to produce sub-micron soot particles with desired particle size and concentration. Different flame conditions were realized by regulating the fuel to air equivalence ratios. Details of the flame selection and flame conditions are provided elsewhere (Pei et al, 2017).

The soot from the propane flame generator was dried with a silica gel diffusion drier, and a Thermo-Denuder (TD) was then used to remove each of the co-produced primary organic components, as shown in Fig. 1. The TD was maintained at 400 °C and the particle residence time within the TD was ~1.6 min. Further downstream, NOx was removed using a NOx denuder filled with charcoal granite (Activated Charcoal Powder USP, Spectrum Chemical Mfg. Corp., USA). The poly-dispersed pure soot cores were charged by a bipolar charger ($^{63}$Ni) and then selected using the first DMA (DMA1, model 3081, TSI Inc., Shoreview, MN, USA, aerosol flow rate 0.4 L min$^{-1}$, and sheath flow rate 6.0 L min$^{-1}$) for size-resolved modulation and subsequent characterization experiments.

### 2.2 Soot modification

Soot particles were modulated using the experimental setup of the Tandem Differential Mobility Analyzer (TDMA) with the integrated flow-tube system and then characterized using the APM, DMAs, and CPC, as shown in Fig. 1. The size selected mono-disperse soot core surface was modified by sulfuric acid coatings using a cylindrical glass mixing chamber (length: 47 cm, diameter: 10 cm) equipped with a temperature-regulated bath system. The modified soot particles were then further modulated by SOA generated via limonene ozonolysis in a low NOx, dark laminar flow tube (flow rate: 1.52 L min$^{-1}$ and effective particle residence: 4.8 min at 25 °C and regulated RH).

Four nominal sizes of soot core, i.e., 75, 100, 150, and 200 nm, were selected using DMA1. The experimental matrix included three major procedural steps: 1) characterization of the pure soot core; 2) modification of the soot core surface with sulfuric acid and characterization of the modulated soot; 3) modulation of the modified soot with limonene ozonolysis products and characterization of the modulated soot.



The acid-induced surface modifications were performed at three different temperatures namely, 1 °C (small=S), 5 °C (medium=M), and 25 °C (large=L), using a temperature-regulated bath system. In the SOA modulation steps, the limonene concentrations in the flow tube were regulated using flowing zero air over limonene in a diffusion vial submersed in a temperature-regulated bath system (Jonsson et al., 2008). The limonene mixing ratios in the flow-tube (56 ppb, 73 ppb, 138 ppb) were precisely regulated by setting the temperatures of the limonene vial bath system to 1 °C (small=S), 5 °C (medium=M), and 15 °C (large=L) at a constant zero air flow rate (0.002 L min⁻¹). Ozone was generated by passing zero air (Ultra Zero Air Generator GT 3000, LNI SCHMIDLIN AG) through a UV-lamp unit (SOG-3, UVP). The ozone concentration used in the experiments was set to a constant value of 1432 μg m⁻³ (~730 ppb) and monitored using a UV photometric ozone analyzer (model 49C, Thermo Environmental Instruments Inc.). A low constant RH (5±1%) was maintained in all cases. The nucleation of limonene ozonolysis products may happen during the modification process; however, since the homogeneous nucleated SOA particles did not have a charged soot core, they cannot be selected by DMA2 and will not influence the soot characterization results.

For (i) pure soot core characterization, the modulation steps involving sulfuric acid and limonene SOA were all bypassed (see Fig. 1), (ii) determining the effect of sulfuric acid on soot mobility and morphological properties, only the modulation steps involving limonene SOA were bypassed (by turning off the ozone production through the UV-lamp power supply; see Table S1 of the supplementary information for the experimental matrix). (iii) determining the effect of pure limonene SOA on soot mobility and morphological properties, the modulation steps involving sulfuric acid were bypassed

## 2.3 Aerosol characterization

### 2.3.1 DMA+CPC for the particle-size number distributions

The post modulation peak diameters of the mono-disperse soot cores were scanned using a combination of a DMA (DMA2, Vienna type, length 280 mm, inner/outer radius 25.0/33.3 mm, aerosol flow rate 2.25 L min⁻¹, and sheath flow rate 10.0 L min⁻¹) and a condensation particle counter (CPC – model 3775, TSI Inc., Shoreview, MN, USA, flow rate 0.3 L min⁻¹), which served as a scanning mobility particle sizer (SMPS).

### 2.3.2 DMA-APM for the size-resolved particle mass

After DMA2, the mass of the mono-disperse particle population was measured using a series combination of an APM (model APM-3600, Kanomax) and a CPC (model 3775, TSI Inc., Shoreview, MN, USA, flow rate: 0.3 L min⁻¹, see Fig. 1). The APM consists of two concentric cylinders rotating at the same angular speed. A voltage was applied over the cylinders and the particles introduced in the gap between these cylinders experienced a centrifugal force. The APM transmitted only





those particles with a specific mass at which the electric force is equal to the centrifugal force. These particles were then counted with a CPC (Rissler et al., 2013). The DMA-APM system was calibrated using two sizes of polystyrene latex spheres (Duke Scientific Corp., USA), in accordance with previously described methodology (McMurry et al., 2002). Moreover, the mass sensitivity of multiply charged particles was determined and for all soot particle sizes, the mass was

overestimated by <5% (see supplementary information).

**2.4 Data analysis**

The particle diameter growth factor, *Gfd*, is calculated from:

$$Gfd = \frac{D_p}{D_{p,0}} ,$$ (3)

where $D_p$ and $D_{p,0}$ are the mobility diameter of particles at a given reaction time and the mobility diameter of the fresh soot

particles, respectively.

The particle effective density $\rho_{eff}$ can be calculated from Eq. (2). The particle organic mass fraction $fm_{org}$ is determined from: $(m_p - m_{SA} - m_0)/m_p$, and the particle sulfuric acid mass fraction $fm_{SA}$ is determined as $m_{SA}/m_p$. Similarly, the material density $\rho_m$ is calculated from the material density of the soot ($\rho_{soot}$: 1.77 g cm$^{-3}$), organic coating ($\rho_{org}$), and sulfuric acid ($\rho_{SA}$) if

volumetric additivity is assumed. The material density of limonene ozonolysis products ($\rho_{org}$: 1.20 g cm$^{-3}$) measured by Chen et al. (2010) is used in this study. This value is similar to the organic aerosol density (1.26±0.04 g cm$^{-3}$) estimated from O:C and H:C determined via SP-AMS measurements performed by Kuwata et al. (2012), and the value (1.3±0.2 g cm$^{-3}$) reported by (Saathoff et al., 2009). The material density of sulfuric acid (sulfuric acid-water mixture) at 5±1% RH ($\rho_{SA}$: 1.84 g cm$^{-3}$) is estimated using the same method employed by Pagels et al. (2009). The material density $\rho_m$ is calculated

from:

$$m_{SA} = m_{p,soot+SA} - m_0 ,$$ (4)

$$fm_{SA} = \frac{m_{SA}}{m_p} ,$$ (5)

$$fm_{org} = \frac{m_p - m_{SA} - m_0}{m_p} ,$$ (6)

$$\frac{1}{\rho_m} = \frac{fm_{org}}{\rho_{org}} + \frac{fm_{SA}}{\rho_{SA}} + \frac{1 - fm_{org} - fm_{SA}}{\rho_{soot}} ,$$ (7)

The mass equivalent diameter $D_{me}$ corresponds to a spherical particle of the same mass, and can be calculated from the particle mass $m_p$ and the material density $\rho_m$:

$$D_{me} = \sqrt[3]{\frac{6m_p}{\pi \rho_m}} ,$$ (8)



The change in particle $D_{me}$ is expressed as the mass equivalent coating thickness $\Delta r_{me}$:

$$\Delta r_{me} = \frac{D_{me} - D_{me,0}}{2} \,,\tag{9}$$

where $D_{me,0}$ and $D_{me}$ are the mass equivalent diameters of the fresh and the coated soot particles, respectively.

The dynamic shape factor $\chi$ can be calculated from the measured mobility diameter $D_p$ and the volume equivalent diameter $D_{ve}$:

$$\chi = \frac{D_p C_{ve}}{D_{ve} C_p} \,,\tag{10}$$

where $C_{ve}$ and $C_p$ are the Cunningham slip correction factors for particles with diameters $D_{ve}$ and $D_p$, respectively. In situ experimental determination of $D_{ve}$ is not currently possible and, hence, soot agglomerates free of internal voids are typically

assumed, leading to $D_{ve} = D_{me}$.

The void space fraction ($F_{vs}$), i.e., volume of voids/total volume of particles derived from the mobility diameter, is calculated from the $D_{me}$ and $D_p$ of fresh and coated soot:

$$F_{vs} = 1 - \frac{D_{me}^3}{D_p^3} \,,\tag{11}$$

In this study, the model developed by Sorensen (2011) is used to describe the relation among the particle mobility diameter $D_p$, primary particle (soot spherules) diameter $d_{pp}$, and number of primary particles (soot spherules) in a soot agglomerate ($N$). We assume that the primary particles of soot agglomerates are in point contact and the material density of these particles is the same as that (1.77 g cm$^{-3}$) of the soot.

$$m_p = N \cdot \frac{\pi}{6} \rho_m d_{pp}^3 \,,\tag{12}$$

$$D_p = d_{pp} N^x, N < 1000 \,,\tag{13}$$

where $x = 1/Dfm = 0.46$ assuming a $Dfm$ of 2.17 for the soot agglomerate. In the original Sorensen model, a fixed exponent $x$ was used for $N < 100$. However, Rissler et al. (2013) suggested that the formula should also be used for $N$ values of up to 1000.

**2.5 Framework for quantifying the morphological transformation**

A framework was developed for quantifying the state of the morphological transformation of soot, i.e., the utilization of material for filling and growth, as shown in Fig. 2. Fig. 2(a) shows the particle diameter growth factor ($Gfd$) as a function of the coating thickness $\Delta r_{me}$ associated with each measurement point. The ideal growth line, denoted by a black solid line,





describes condensation of material on a perfect incompressible solid sphere with the same initial mobility diameter and mass as a fresh soot particle. We hypothesize that when material condenses on the soot agglomerate, the growth process will be described by $Gfd$ parallel to the ideal growth line, while void filling will be described by a line parallel to the $x$-axis. The coating thickness for void filling ($\Delta r_{me,f}$) and the coating thickness for particle growth ($\Delta r_{me,g}$) are indicated by the red arrow

(parallel to the $x$-axis) and the green arrow (parallel to the ideal growth curve), respectively. From this framework, the fraction of void space filled ($F_{vs,f}$), volumes of material utilized for void filling ($V_f$) and diameter growth ($V_g$), and the percentages of material utilized for filling ($P_f$) and growth ($P_g$) can be derived as follows (based on the assumption of a concentric core shell structure; see Fig. 2(b) for illustration):

$$F_{vs,f} = \frac{(D_{me,0} + 2\Delta r_{me,f})^3 - D_{me,0}{}^3}{D_p{}^3 F_{vs}} , \tag{14}$$

$$V_f = \frac{\pi}{6}\left[(D_{me,0} + 2\Delta r_{me,f})^3 - D_{me,0}{}^3\right] , \tag{15}$$

$$V_g = \frac{\pi}{6}\left[(D_{me,0} + 2\Delta r_{me,f} + 2\Delta r_{me,g})^3 - (D_{me,0} + 2\Delta r_{me,f})^3\right] , \tag{16}$$

$$P_f = \frac{V_f}{V_f + V_g} , \tag{17}$$

$$P_g = \frac{V_g}{V_f + V_g} , \tag{18}$$

During the experiment, the soot undergoes step-wise morphological transformation (see section 3.3 for details). The fraction of internal voids ($F_i$) and the fraction of open voids ($F_o$) in the soot agglomerate are determined based on the following hypotheses: (i) in the case of SOA, the open voids in the condensed material are preferentially filled prior to the onset of growth, and (ii) at the onset of growth, $F_o$ is equal to the fraction of void space filled ($F_{vs,f}$), whereas $F_i = 1 - F_o$. The volume equivalent diameter including internal voids ($D_{ve,i}$) is then given as:

$$D_{ve,i} = \sqrt[3]{D_{me}{}^3 + D_p{}^3 F_{vs} F_i} , \tag{19}$$

## 3 Results

### 3.1 Fresh soot properties

As previously mentioned, fresh pure soot is obtained by heating the soot particles in the TD at 400 °C to remove co-produced organics from the soot generator. For the soot transformation study, four fresh pure soot particle sizes with nominal

mobility diameters ($D_p$) of 75, 100, 150, and 200 nm are selected using DMA1. The corresponding actual mobility diameters (actual $D_p$) are scanned using DMA2 and CPC, as shown in Fig. 1. The $D_p$, mass equivalent diameter ($D_{me}$), mass ($m_p$),



effective density ($\rho_{eff}$), and the dynamic shape factor ($\chi$) of the particles are listed in Table 1. As the table shows, the final measured $D_p$ of the modified soot differs only ~5% from the instrumental accuracy of the DMA1 and the SMPS (DMA2+CPC) system. The calculated $D_{me}$ is smaller than the $D_p$ corresponding to each size, i.e., $D_{me}$ of 57.1, 70.8, 95.3, and 116.2 nm are obtained for $D_p$ of 75, 100, 150, and 200 nm, respectively. This suggests that the soot particle contains

significant fractions of void space ($F_{vs}$). In fact, the estimated fraction of void space within a fresh soot particle (listed in Table 1) is fairly large, e.g., $F_{vs}$ values of 58%, 63%, 72%, and 78% are obtained for $D_p$ of 75, 100, 150, and 200 nm, respectively. These void fractions account for both internal and external voids (see section 3.4). However, the corresponding dynamic shape factor ($\chi$), a parameter describing the sphericity of the particle (see Table 1), is estimated based on the assumption of no internal voids. The value of $\chi$ increases from 1.66 to 2.29 with increasing (from 75 to 200 nm) $D_p$. This

indicates, as reported in previous studies (Xue et al., 2009; Khalizov et al., 2013), that the shape irregularity of the particle increases with increasing mobility diameter. However, compared with previous studies, our quantification considers internal voids (which in fact results in significantly smaller $\chi$) and therefore yields more accurate $\chi$ (see section 3.4).

The morphological structure of the soot agglomerate may also be described via the mass-mobility exponent ($Dfm$), which

characterizes the primary particles (spherules) and describes their arrangement within the agglomerate. Averaged over all four sizes, i.e., 75, 100, 150, and 200 nm, a $Dfm$ of 2.28 is derived by fitting the actual $D_p$ and the soot particle mass. This value is consistent with that reported for fresh soot particles from a diffusion propane burner (Pagels et al., 2009). We used the $Dfm$ of pure BC particles in Eq. (11) and Eq. (12) to determine the size of the primary spherules ($d_{pp}$), and a value of ~28 nm is obtained for all four sizes of the soot agglomerates. The $d_{pp}$ reported in previous studies vary significantly and are

typically lower than the value obtained in this study. For e.g., Pagels et al. (2009) and Zhang et al. (2008) obtained a $d_{pp}$ of ~15 nm via TEM image analysis. Similarly, Rissler et al. (2013) obtained, via TEM analysis, values of 24, 27, 28, 11, and 13 nm, for particles (geometric mean diameter: 50 nm) associated with a heavy-duty transient diesel engine, heavy-duty idling diesel engine, light-duty idling diesel engine, candle, and propane flame soot, respectively. Our $d_{pp}$ (i.e., 28 nm) is similar to that obtained from field measurements performed by Rissler et al. (2014), indicating that the soot agglomerates considered in

this study are similar to real-world soot particles. According to the Sorensen model, the $d_{pp}$ determines the void fraction in the agglomerate. Therefore, quantifying the $d_{pp}$ is essential for a complete description of the morphological state of the soot agglomerate.

Generally, comparison with previous studies is difficult, owing to the use of different fresh soot material as the substrate for

subsequent soot transformation studies. The properties reported in selected studies are summarized in Table S4. As the table shows, fresh soot particles with similar mobility diameter ($D_p$ ~100 nm) have similar $Dfm$ (2.14–2.28), but widely varying (15 nm–45 nm) $d_{pp}$. This indicates that the soot agglomerates are formed through a similar coagulation process after the nascent primary spherules are formed in the flame. The results reported in the literature can be classified into two groups: (1) Pagels et al. (2009), Xue et al. (2009), Qiu et al. (2012), Khalizov et al. (2013), and Peng et al. (2016) generated similar soot



with $\chi$ of ~2.2 and $d_{pp}$ of 15–21 nm; (2) Guo et al. (2016) and Ghazi and Olfert (2013) generated similar soot with relatively compact morphology characterized by $\chi = 1.5$ and $d_{pp} = 45$ nm. The compactness of the soot considered in this study ($\chi = 1.81$ and $d_{pp} = 28$ nm) represents an intermediate of these two groups. This results from the fact that, for any given soot core mobility diameter, the fractality of the soot agglomerate increases with decreasing $d_{pp}$. In previous studies, $\chi$ was calculated

based on the assumption that internal voids were absent from the soot agglomerate. This assumption yields $D_{me} = D_{ve}$. In this work, $D_{ve}$ increases whereas $\chi$ decreases with the occurrence of internal voids (see sections 3.2 and 3.3 for the experimentally determined open-void fractions and relevant discussions).

**3.2 Formation of limonene ozonolysis SOA**

Limonene SOA was formed on the soot surface regulated by VOC, ozone and sulphuric acid concentrations in the

experiments. In general, compared with those formed from the reactions of VOC and ozone, significantly higher amounts of SOA formed when the reactions occurred on the acid-coated soot surfaces. The levels of acidity on the surface play an important role in this formation. Fig. 3 shows the masses of different levels of sulfuric acid and SOA coatings associated with a 200 nm soot core. The amounts of sulfuric acid and limonene SOA coatings on a soot particle are classified as small (S), medium (M), and large (L). The mixed coating experiments cover the full matrix, i.e., BC coated with each of the S, M,

and L amounts of sulfuric acid is also co-coated with S, M, and L amounts of limonene SOA. Soot cores of other sizes (75, 100, and 150 nm) exhibit similar trends to those observed for the 200 nm soot. Acidity-induced enhancement of the SOA mass is attributed to acid-catalyzed heterogeneous reaction mechanisms (such as hemiacetal and acetal formation) of carbonyls in SOA, polymerization, and aldol condensation (Jang et al., 2002). The mass of SOA formed on the BC particle surface under L SOA and L sulfuric conditions was significantly higher than the total mass of L SOA formed under L

sulfuric acid only and on the acid-free surface. Therefore, we achieved various particle coating thicknesses, which allowed a thorough investigation of the morphological transformation of the BC particle.

**3.3 Growth of the particle mobility diameter**

The mobility diameter growth factor ($Gfd$) implies that the morphological transformation of soot agglomerates results from condensation of the material. Fig. 4(a)–(d) shows $Gfd$, associated with initial fresh soot mobility diameters of 75, 100, 150,

and 200 nm, as a function of the mass equivalent coating thickness ($\Delta r_{me}$). The $Gfd$ values are measured for the full experimental matrix described above. In Fig. 4, the ideal growth curve is denoted by the black dashed line and the pie chart denoting each data point shows the chemical composition of fresh and processed soot particles. The black, red, and green colors in each pie chart represent the mass fraction of black carbon, sulfuric acid, and organics, respectively. We hypothesize that the morphological transformation of soot particles result from the: (i) filling of void space within a fractal soot

agglomerate; (ii) growth of mobility diameter identical to that of the ideal sphere (parallel to black dashed line); (iii) rearrangement of primary spherules, i.e., collapse, within a fractal soot agglomerate due to surface forces exerted by the





condensed material. Condensational growth of the material will increase the mobility size and mass of the particles. Adding material to the void space of the soot structure will increase the mass, but the mobility size will remain the same or decrease due to collapse. We have observed all three effects (see Fig. 4). The morphological transformation of the soot fraction results from a combination of two or more of these processes and, when expressed as the *Gfd*, exhibits a strong particle size

dependence. The morphological properties associated with the four sizes of fresh soot are shown in Table 1. Furthermore, the *Gfd* associated with different coatings are illustrated by purple lines and the trends are summarized as follows, the: (i) line parallel to the ideal sphere growth line (dashed) indicates that all material added was utilized for growth of the particle diameter; (ii) line parallel to the *x*-axis indicates that all condensed material was utilized for filling the void space within a fractal soot agglomerate; (iii) negative slope indicates a combination of void space filling and collapse of the soot particle.

The soot structure of fresh soot particles with diameter and mass of 75 nm and 0.17 fg (Exp. 1), respectively, is relatively resistant to collapse. The major morphological transformation occurs sequentially in a step-wise manner, with filling of void space (shown by the solid purple line parallel to the *x*-axis) and growth of the diameter (solid purple line parallel to the black dashed line) as a function of $\Delta r_{me}$ (mass equivalent coating thickness; see Fig. 4(a)). When the heaviest coating is applied to

the 75 nm soot agglomerate with L sulfuric acid (0.26 fg) + L SOA (2.37 fg) (Exp. 15), 94% of the total volume of the condensed material is utilized for diameter growth (see Fig. 4(a)). Only 6% is utilized for filling the void space, but this 6% has filled 97% of the void space within the soot particle (Table 2). The morphological evolution of the soot appears to be independent of the type of condensed material. In other words, in the present study, pure sulfuric acid, pure SOA, and acidity-mediated SOA participate indiscriminately in the process of void filling and growth of the 75 nm soot particle. In the

(i) S and M pure sulfuric acid experiments (Exp. 4, 0.02 fg and Exp. 8, 0.03 fg, respectively), 93% and 100%, respectively, of the condensed sulfuric acid are utilized for filling. (ii) L pure sulfuric acid experiment (Exp. 12, 0.26 fg), 23% of the condensed material is utilized for void filling, resulting in the filling of 25% of the voids. (iii) S and M pure SOA (Exp. 2, 0.08 fg and Exp. 3, 0.11 fg, respectively), only 23% and 35% of the condensed material are utilized for void filling, resulting in 12% and 26%, respectively, of filled voids. These results show that the growth of material on the soot plays a dominant

role in soot transformation. We assume that the microphysical structure of the soot agglomerate considered in this study contains mainly two types of voids (as outlined by DeCarlo et al. (2004)) namely, (i) internal voids, i.e., space shielded by soot primary spherules, and (ii) open voids, i.e., space open to the atmosphere (shown in Fig. 2(c)). We assume that both types of voids lie within a hypothetical sphere of mobility diameter. Furthermore, we interpret our data in light of the knowledge that the open voids will be preferentially filled over the internal voids, due to the shielding of primary spherules.

This preferential filling characterizes the microphysical transformation of the soot considered in this work. The space within the internal voids is probably bottlenecked due to the narrow opening, thereby preventing the entry of material except for large masses (5–15 times of the soot particle mass) of the condensed material (as observed in Exp.11 and Exp. 13–15). These occurrences seem to control the step-wise filling and subsequent growth process, which proceed until the voids are all filled. The indiscriminate behavior of pure sulfuric acid and pure SOA in the morphological transformation of 75 nm soot particles



is attributed to the same growth and filling patterns (see Fig. 4(a)). This suggests that most of the void space is occupied by internal voids, as evidenced by the filling of only 10% of voids prior to the onset of sharp growth. The percentage of voids filled and the growth of the particle diameter are quantified (see Table S2) at each step shown in Fig. 4(a). Thus, we have quantified the state of morphological transformation (from beginning to end) as a function of the amount and type of

material condensed on the soot agglomerate.

We also consider 100 nm soot consisting of particles with mass of 0.33 fg. Upon condensation of the material, the major morphological transformation occurs sequentially in a step-wise fashion, i.e., filling of void space and growth of diameter, as previously described for the 75 nm soot (see Fig. 4(b)). When condensed L sulfuric acid (0.34 fg) + L SOA (1.50 fg) is

applied (Exp. 31), 90% of the total volume of the condensed material is utilized for diameter growth. The remaining 10% is utilized for filling, resulting in filling of 44% of the void space. In contrast to that observed for the 75 nm soot, the response on the morphological transformation of the 100 nm fresh soot exhibits a strong dependence on the type of material condensed. Most of the volume (54%) of the L pure sulfuric acid (Exp. 28) is utilized for initial filling (up to 10 nm $\Delta r_{me}$ filled 30% of the void space). For L pure SOA (Exp. 19), only 17% of void filling occurs, but the diameter growth is greater

(80% of material utilized for growth and only 20% for filling) than in the case of L pure sulfuric acid. This eventually leads to two different pathways of morphological transformation of the soot particle, leading to a hysteresis of transformation (see Fig. 4(b) and Table S2 of the supplementary information). This hysteresis constitutes a response to the nature of condensing material and the type of microphysical void space (internal voids or open voids) within the soot particle. For e.g., the condensation of M (Exp. 24) followed by L (Exp. 28) sulfuric acid on pure soot lead to a void filling of 30%, due to the

higher surface tension relative to that associated with L SOA. This higher tension yields efficient filling of some (21%) of the internal voids prior to the onset of major particle growth. Similar forces and similar filling (26%) can be achieved with additional L SOA and S sulfuric acid (Exp. 23) on the soot by increasing the amount of material. The type of growth and void filling associated with the hysteresis suggest that the 100 nm soot agglomerate contains mainly internal voids, which require a considerable amount of condensable material to fill. For example, only 44% (maximum) of void space is filled

when almost five times the soot core mass consisting of acidity-mediated SOA (1.84 fg) is condensed (Exp. 31). This suggests that, in the case of pure SOA, the internal void structure inhibits void-space filling (growth begins at 10% of void-space filling). However, 30% of void filling is achieved for L sulfuric acid (0.34 fg, Exp. 28). Morphological transformation from the L sulfuric acid experiment (Exp. 28) to L sulfuric acid + L SOA experiment (Exp. 31), is dominated by growth of the diameter (the extent of this growth ranges from 7% to 59% relative to void filling, i.e., 30% (L sulfuric acid) to 44% (L

sulfuric acid + L SOA)). Owing to the surface tension of sulfuric acid, its vapors can efficiently enter the narrow spaces of internal voids within the soot agglomerate, thereby resulting in filling prior to the onset of growth. However, in the case of SOA, growth begins with complete filling of the open voids, indicating that the initial diameter growth (leading to 10% of void filling) determined the open-void fraction (i.e., 10%).



In the case of 150 nm fresh soot with a mass of 0.75 fg, 76% of the total volume of the condensed material is utilized for diameter growth. In contrast, 24% is utilized for filling when a mixture of L sulfuric acid + L SOA (Exp. 46) is condensed, leading to 39% of void-space filling and 30% diameter growth. The major morphological transformation occurs sequentially in a step-wise manner (see Fig. 4(c)), i.e., with sequential filling and growth of the diameter. Furthermore, the response to

the nature of the material and the hysteresis of the transformation differ significantly from those of the 100 nm particle. For example, 2% growth is initially achieved in three experiments namely, M sulfuric acid (Exp. 39), L sulfuric acid (Exp. 43), and S SOA (Exp. 33). Void filling of 3%, 25%, and 9%, respectively, are achieved for these experiments. Increasing the amount of SOA, e.g., in Exp. 39 (M sulfuric acid) and Exp. 43 (L sulfuric acid), yields a significant change in the response to morphological transformation pathways, although similar growth is achieved. In other words, additional L SOA in Exp. 39

(Exp. 42) leads to 13% growth and 29% void filling, while additional S SOA to L sulfuric acid in Exp. 43 (Exp. 44) leads to 11% growth and 42% void filling. This leads to hysteresis of the transformation, as shown in Fig. 4(c). The process of void filling and particle growth is quantified at each step shown in Fig. 4(c) (see Table S2 of the supplement). The same level of growth (i.e., 2%) occurs in these three experiments irrespective of the nature of the condensed material and the percentage of the volume of the condensed material that is utilized for void filling (39%, 81%, and 70% for Exp. 39, Exp. 43, and Exp. 33,

respectively). This suggests that most of the initial filling occurs for open voids. When open voids are filled completely, the filling of internal voids occurs preferentially to growth by high surface tension species, such as pure sulfuric acid (e.g., 56 Nm$^{-1}$). SOA with low surface tension (e.g., 25–30 Nm$^{-1}$) will, however, lead to growth. Compared with that required for growth of the diameter (high surface energy barriers), a smaller surface area is required for the filling of voids (lower energy barriers). Sulfuric acid tends to fill even the internal voids prior to growth of the diameter, but SOA fills these voids only

after growth has begun. In other words, the ability of sulfuric acid to fill some internal voids prior to growth can be attributed to its high surface tension and low vapor pressure. Owing to these characteristics, the relatively low surface energy barriers of the internal voids are easily overcome. The filling of open voids will largely be completed when SOA growth begins. This hypothesis is verified by quantifying the fraction of internal to open voids within the boundary of the hypothetical sphere of mobility diameter. As in the case of the 75 nm and 100 nm soot agglomerates, the 150 nm soot contains a smaller fraction

(i.e., 10%) of open voids than internal voids. All (100%) of these voids are filled, whereas only 33% of the internal voids (constituting 90% of all voids in the soot) is filled during the heaviest coating experiment (Exp. 46). The morphological transformation process of the 150 nm soot differs (e.g., from the viewpoint of hysteresis) from that of the 75 nm and 100 nm agglomerates. For e.g., during the heaviest coating experiments, 6%, 10%, and 24% of the condensed material are utilized for filling in the 75 nm, 100 nm, and 150 nm soot, respectively. However, in terms of absolute fractions, condensational

growth still dominated over void filling for both sulfuric acid and SOA.

200 nm soot particle with a mass of 1.46 fg is also considered. When coated with L sulfuric acid (0.68 fg) + L SOA (2.00 fg) (Exp. 62), 42% of the total volume of condensed material is utilized for diameter growth and 58% is used for filling. This filling resulted in 41% of the void space filled. The percentage of condensed material utilized for filling increases



significantly when the soot size is increased from 75 nm to 200 nm, confirming that more void space occurs in the larger size soot. This is consistent with the estimated $F_{vs}$ result. As anticipated, the major morphological transformation results from sequential step-wise filling and growth of the 200 nm soot particle. Furthermore, the response to the nature of the condensed material and the hysteresis of the transformation are similar to those of the 150 nm soot particle. Internal arrangement of

primary spherules within the hypothetical sphere of mobility diameter that shrinks or collapses ($Gfd < 1$) is also possible. Upon the addition of S/M/L sulfuric acid (Exp. 51, Exp. 55, Exp. 59) and S SOA (Exp. 48) to the 200 nm soot agglomerate, indiscriminate shrinkage occurs, owing to the rearrangement of primary spherules. The fraction of open voids may increase with this rearrangement and shrinkage of 4% and 2% occur for L sulfuric acid (Exp. 59) and SOA (Exp. 48). The initial fraction of open voids (10%) is filled by S SOA (Exp. 48), and additional (7%) open voids are created with subsequent

condensation of the material (Exp. 48). These additional voids are filled during Exp. 56 and Exp. 57 (as indicated by negligible growth of the diameter). However, compared with SOA, sulfuric acid is more efficient at filling the internal voids when more material is condensed (Exp. 59–62) and 31% of these voids (constituting a void fraction of 83%) is filled. This filling accounts for 26% of the total void space. Detailed estimates associated with each step of filling and shrinkage of the 200 nm soot are provided in Table S2 of the supplement.

Previous studies considering the morphological transformation of soot using similar techniques are also evaluated with the framework described above (Guo et al. 2016; Qiu et al. 2012). Fig. 5 shows the $Gfd$ as a function of $\Delta r_{me}$ for the 100 nm soot transformation measured in a smog chamber by Guo et al. (2016) and Qiu et al. (2012). The major difference between those experiments and the experiments employed in this study is that a flow tube (rather than a smog chamber) is used in the

present work. The smog chamber allows dynamic particle growth, i.e., particle properties including mobility diameter and mass are measured in 40-min intervals while particles are continuously growing. This may introduce some shift in the mapping of the mass and mobility size, leading to overestimation of $\Delta r_{me}$. Nevertheless, the framework provides a fairly good depiction of the features associated with the morphological transformation (i.e., filling and growth) of their soot. These features are consistent with the ideal sphere growth theory. Our framework reveals that their soot is composed mainly of

open voids (84–95%) as indicated by their growth curve, where the growth line is parallel to the $x$-axis (no growth). In their studies, growth and filling occur sequentially, but over very short time intervals, indicating that internal voids are relatively easy to fill. This highlights the major differences (see Table 3) between the present study and previous studies. These results show that the framework developed in this study is quite capable of evaluating the mass and mobility data required for interpreting the morphological transformation of soot of various sizes.

**3.4 Dynamic shape factor**

The dynamic shape factor $\chi$ is an important parameter used to represent the shape of both fresh and coated soot particles. $\chi = 1$ and $\chi > 1$ correspond to perfectly spherical particles and irregularly shaped particles, respectively. Fig. 6(a)–(d) shows $\chi$ as





a function of the coating thickness $\Delta r_{me}$ for four different fresh soot sizes and the same soot cores coated with either pure sulfuric acid, pure SOA or acidity-mediated SOA. These are calculated from Eq. (10) assuming that the soot agglomerate is free of internal voids. However, as previously stated, our experimental results show that internal voids dominate the total void space in all four cases. Void fractions of 88%, 91%, 89%, and 83% are obtained for the 75 nm, 100 nm, 150 nm, and

200 nm soot, respectively. Therefore, the $\chi$ values obtained based on the *no internal voids* assumption differ significantly (see Fig. 6) from the experimentally determined values obtained in this study. This assumption neglects the *in-situ* morphology of the soot agglomerate, and stipulates (based on the notion of a void-less sphere) that the equivalent volume is equal to the sum of the all primary spherules. These findings highlight the serious shortcomings of the assumption that $D_{ve} = D_{me}$ and the implications for atmospheric surface processes that are considered critical for modeling-based studies. As shown

in Fig. 6(a)–(d), a significant amount of material is condensed on the soot particles, but a perfect sphere remains elusive. For e.g., the thickness of the coating on the 75 nm particle is at least two times larger than the initial mobility diameter, but $\chi$ still deviates from unity (Exp. 15). This results from the fact that $\chi$ is estimated (Eq. 10; $D_{ve} = D_{me}$) based on the assumption that the internal voids measured in our experiment are all open voids. The $\chi$ values determined in previous studies (see Table S4) based on the *no internal voids* assumption appear to be consistent with each other for the wrong reason. In reality, and as

confirmed in this study, the occurrence of internal voids in a soot agglomerate is unavoidable. Therefore, we suggest that the framework introduced in this work should be developed using an experimental setup, i.e., a flow tube integrated with a DMA-APM. This setup will yield *in-situ $D_{ve}$* associated with the morphological characteristics of soot and the transformation of these characteristics upon the condensation of material.

### 3.5 Effective density

Fig. 7 (a)–(d) shows the effective density ($\rho_{eff}$) as a function of the coating thickness ($\Delta r_{me}$) for four different fresh soot sizes. Ideal growth lines of effective density correspond to growth of material on a perfect sphere with the same mass and mobility diameter as fresh soot grown by either pure sulfuric acid or pure SOA. The coating thickness of the ideal sphere is based on the corresponding mass-equivalent fresh soot.

As described in section 3.3, the morphological transformation (mostly the growth) responds indiscriminately to the addition of pure sulfuric acid and pure SOA to the 75 nm soot. The same response occurs for the effective density of the coated particles. For example, the pure sulfuric acid points are consistent with the ideal growth curve shown in Fig. 7(a) and most of the SOA and acidity-mediated SOA points lie on the ideal growth curve of SOA. These observations support our description of the morphological transformation of the 75 nm soot (see section 3.3). Unlike those associated with the 75 nm soot, the

sulfuric acid as well as the SOA and acidity mediated SOA points of the 100 nm soot lie slightly above the respective ideal growth curves of pure sulfuric acid and SOA (see Fig. 7(b)). This results from the fact that more material is utilized for void filling (than for diameter growth), leading to an increase in the particle mass without diameter growth (see section 3.3 for





details of the 100 nm particle). The increase in mass results in a sharp increase in $\rho_{eff}$. However, when particle growth occurs, $\rho_{eff}$ increases only modestly and follows (for the most part) the ideal curve. In the case of acidity-mediated SOA, most of the points lie between the ideal growth curves of two pure substances, indicative of combined sulfuric acid and SOA growth as well as filling. The 100 nm particle transformation is still dominated by growth and, hence, points from the acidity-mediated

SOA heavy coating experiment are close to ~1.1 g m$^{-3}$ (see section 3.3), which approaches the pure SOA ideal curve. For the 150 nm soot, L sulfuric acid points lie above the sulfuric acid ideal growth curve, which describes the response to filling. L sulfuric acid + S/M/L SOA points result from filling and growth, as discussed in section 3.3. However, the corresponding $\rho_{eff}$ values evolve in a complex manner as the materials (pure soot, pure sulfuric acid, and SOA) in the system have different material densities and mass fractions. The $\rho_{eff}$ increases moderately in these experiments when coatings from S to L SOA are

applied. Other points, representing mainly the growth-dominant transformation of the 150 nm soot via SOA condensation, lie close to the SOA ideal growth curve, as shown in Fig. 7(c). In the 200 nm soot, most of the initially condensed material is utilized for filling and growth is negligible. Therefore, points from the L sulfuric acid experiment lie above the sulfuric acid ideal growth curve. Similarly, several points from the SOA coating experiment lie above the SOA ideal growth curve. The occurrence of points above the curves is consistent with morphological transformation linked to the filling of voids. The

morphological transformation occurring during the heavy acidity-mediated SOA coating experiments (Exp. 60–62) is also dominated by the filling of voids. This filling yields increased $\rho_{eff}$ for points well above the SOA ideal growth curve and around the sulfuric acid ideal growth curve. In each case, $\rho_{eff}$ results from the microphysical transformation of the soot agglomerate and changes in this value are consistent with the findings described in section 3.3.

## 4 Conclusion

Soot particles generated from a propane flame were aged via condensation of sulfuric acid, limonene ozonolysis products or a mixture of both, in a laminar flow tube system. To the best of our knowledge, this is the first study that considers the effect of coatings with two chemical components (i.e., sulfuric acid and SOA) on soot morphology. A framework is developed for quantifying the microphysical transformation of soot on the condensation of material. This framework utilizes experimental data and the hypothesis of ideal sphere growth and filling of voids to quantify the distribution of condensed materials during

a two-step process consisting of diameter growth and void filling. Using this framework, we quantify the percentage of material that is utilized for particle growth and void filling at each step. In the initial stages, filling is the dominant process followed by some growth, which leads to the accumulation of sufficient material. This material exerts a large surface force that facilitates further filling. The process continues in several sequential steps depending on the initial morphology of the fresh soot and the nature as well as the amount of condensed material. Using the same framework, we estimate the fraction

of internal voids and open voids and use this information to derive the volume equivalent diameter of a soot agglomerate containing internal voids. The *in-situ* dynamic shape factor is also calculated from the parameters derived by the framework. The dynamic shape factor estimated from traditional assumptions and methods differs significantly from the value



determined in this study. In fact, most of the fresh soot particles considered in this study are largely spherical, with a dynamic shape factor of ~1.1. The effective density is strongly correlated with the morphological transformational responses to the condensed material on the soot particle and the resultant effective density is determined by the (i) nature of the condensed material and (ii) morphology and size of the fresh soot.

This work represents the first study that quantifies (*in-situ*) microphysical changes in soot morphology, providing details of both the fresh and coated soot particles. The employed framework may be useful for developing a model that determines the morphological transformation and microphysical properties (including hygroscopic and optical properties) of soot in the atmosphere.

**Acknowledgement**

This work was supported by the Swedish Research Council for Environment, Agricultural Sciences and Spatial Planning (FORMAS: Project No. 214-2011-1183), ModElling the Regional and Global Earth system (MERGE) and Climate, Biodiversity and Ecosystem services (ClimBEco).

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



**Table 1. Mobility diameter, mass equivalent diameter, mass, effective density, dynamic shape factor, void space fraction, primary particle size, and number of primary particles of different sizes of fresh soot particles (errors: standard deviation of all the values).**

| Nominal $D_p$ (nm) | Actual $D_p$ (nm) | $D_{me}$ (nm) | mass (fg) | $\rho_{eff}$ (g cm$^{-3}$) | $\chi^{\#}$ without internal voids | $\chi$ with internal voids | $F_{vs}$ (%) | $d_{pp}$ (nm) | $N$ |
|---|---|---|---|---|---|---|---|---|---|
| 75 | 76.0±1.4 | 57.1±0.2 | 0.17±0.01 | 0.77±0.08 | 1.66±0.03 | 1.47 | 58 | 27.9 | 8 |
| 100 | 99.0±1.1 | 70.8±1.2 | 0.33±0.02 | 0.65±0.07 | 1.81±0.04 | 1.55 | 63 | 28.5 | 15 |
| 150 | 146.2±1.4 | 95.3±1.1 | 0.77±0.03 | 0.48±0.04 | 2.10±0.03 | 1.60 | 72 | 27.7 | 39 |
| 200 | 193.3±3.4 | 116.2±4.2 | 1.46±0.16 | 0.39±0.09 | 2.29±0.09 | 1.92 | 78 | 28.0 | 72 |

$^{\#}$ these dynamic shape factors were calculated assuming that internal voids were absent from the soot agglomerate

**Table 2. Quantification of volume of condensed material utilized for filling and growth of the mobility diameter when coated with L amount of sulfuric acid and L amount of SOA. Results of other cases are shown in Table S2 of the supplement.**

| Soot core $D_p$ (nm) | Soot Core $F_{vs}$ (%) | $\Delta r_{me}$ of coated particle (nm) | $\Delta r_{me}$ for filling (nm) | $\Delta r_{me}$ for growth (nm) | Volume of total coating material (×10$^5$ nm$^3$) | Volume of material used for filling (×10$^5$ nm$^3$) | Volume of material used for growth (×10$^5$ nm$^3$) | Volume of soot core void space (×10$^5$ nm$^3$) | Fraction of void space filled (%) | % of material went to filling | % of material went to growth | Fraction of internal voids (%) | Fraction of open voids (%) |
|---|---|---|---|---|---|---|---|---|---|---|---|---|---|
| 75 | 58 | 52.1 | 9.3 | 42.8 | 21.00 | 1.30 | 19.70 | 1.33 | 97 | 6 | 94 | 90 | 10 |
| 100 | 63 | 36.9 | 7.3 | 29.6 | 13.90 | 1.40 | 12.50 | 3.20 | 44 | 10 | 90 | 91 | 9 |
| 150 | 72 | 35.2 | 12.4 | 22.8 | 19.29 | 4.54 | 14.75 | 11.78 | 38 | 24 | 76 | 91 | 9 |
| 200 | 78 | 30.5 | 20.5 | 10.0 | 20.92 | 12.13 | 8.79 | 29.50 | 41 | 58 | 42 | 83 | 17 |

10    **Table 3. Literature values reported for fraction of void space, internal voids, open voids, and void space filled at $\Delta r_{me}$ = 20 nm compared with the values obtained in this study.**

| | Fraction of void space, $F_{vs}$ (%) | Fraction of internal voids, $F_i$ (%) | Fraction of open voids, $F_o$ (%) | Fraction of void space filled, $F_{vs,f}$ (%) at $\Delta r_{me}$ = 20 nm |
|---|---|---|---|---|
| Qiu et al. (2012) | 76 | 16 | 84 | 126 |
| Guo et al. (2016) | 77 | 5 | 95 | 169 |
| This study | 63 | 91 | 9 | 18 |



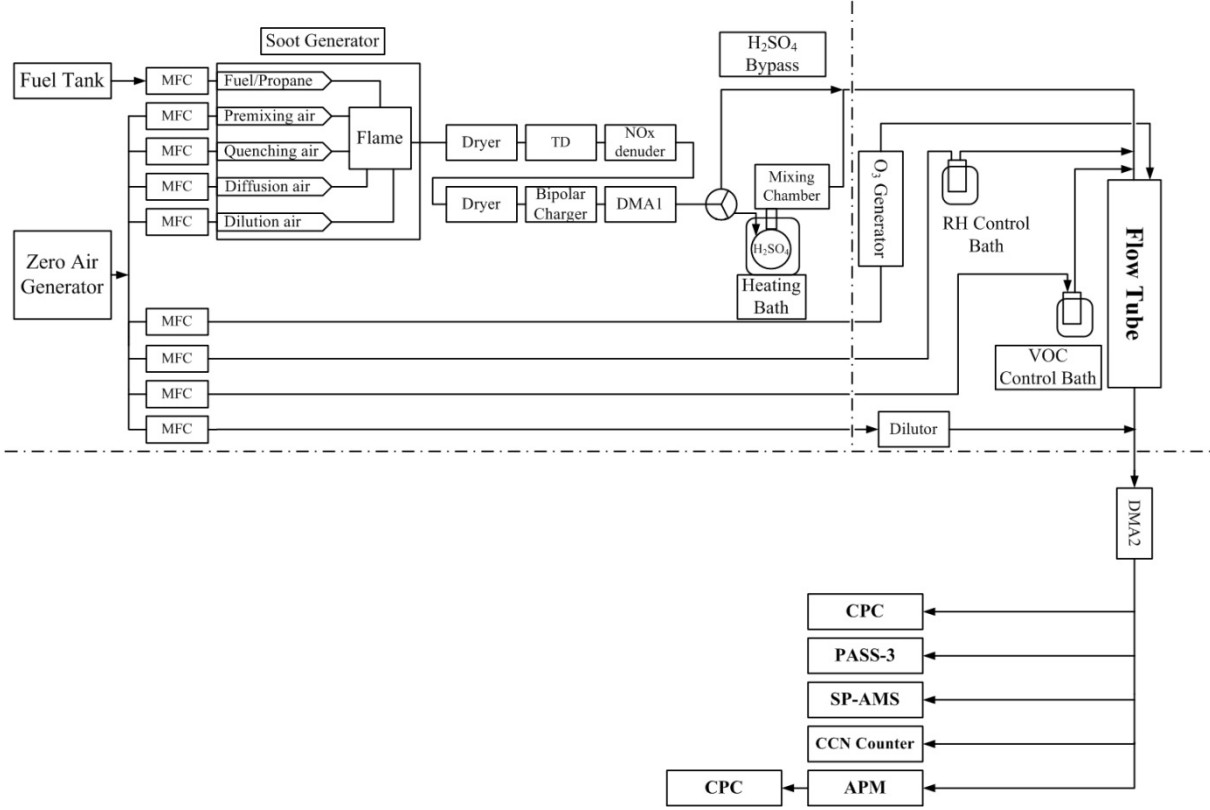

**Figure 1. Schematic of the experimental setup.**

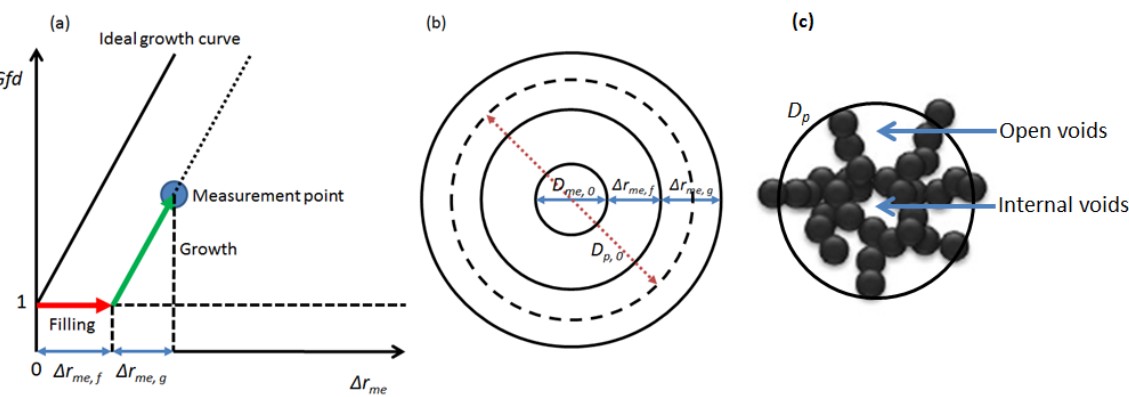

5    **Figure 2. Illustration of the framework for quantifying the morphological transformation and structure of soot agglomerate.**



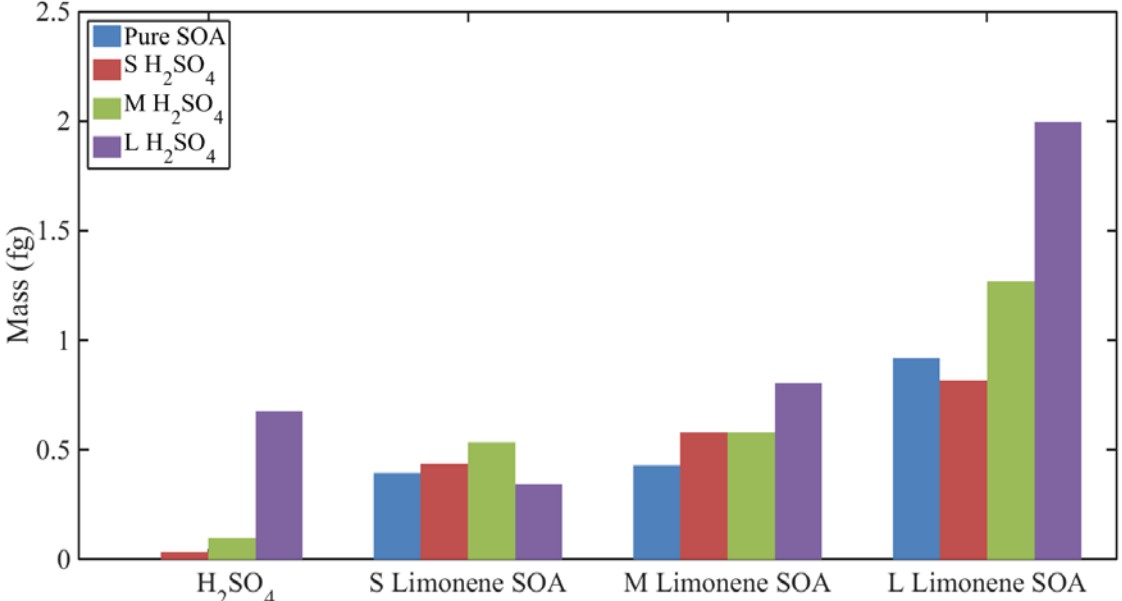

**Figure 3. Mass associated with different levels of sulfuric acid and SOA coatings for the 200 nm soot core.**




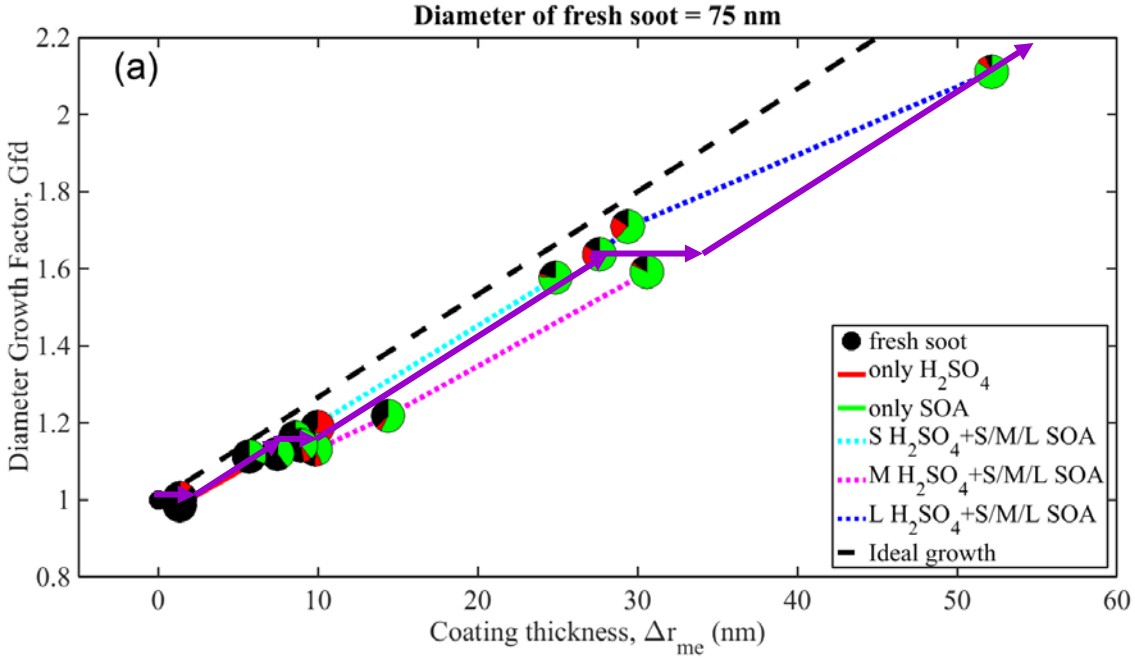

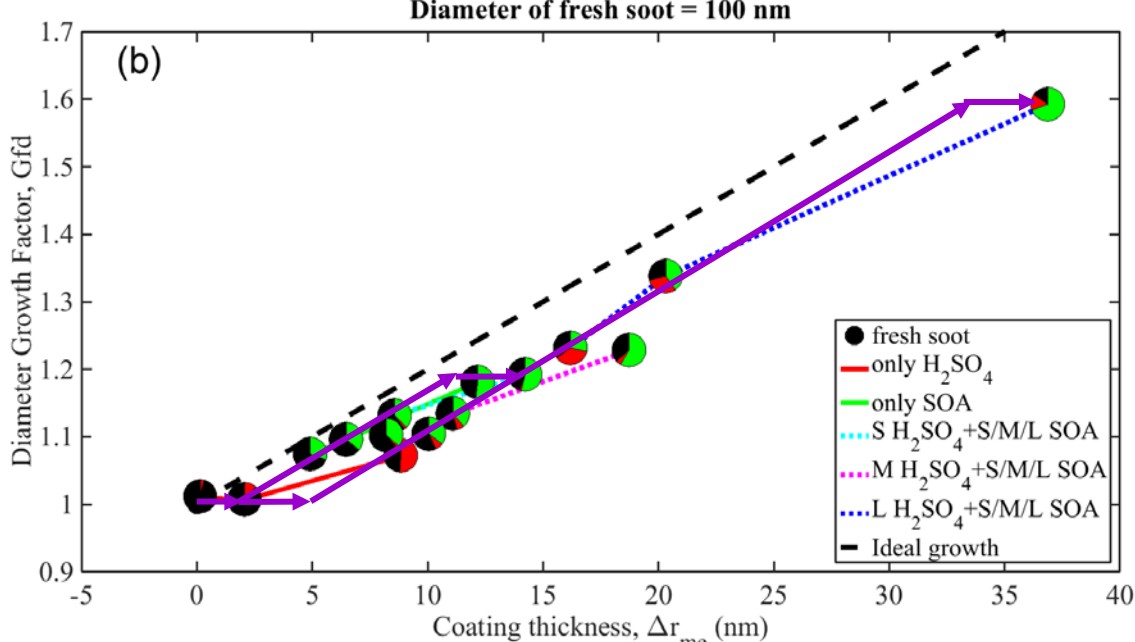





**Figure 4. Mobility diameter growth factor (*Gfd*) with initial fresh soot mobility diameter of 75, 100, 150, and 200 nm as a function of the mass equivalent coating thickness (*Δr$_{me}$*).**



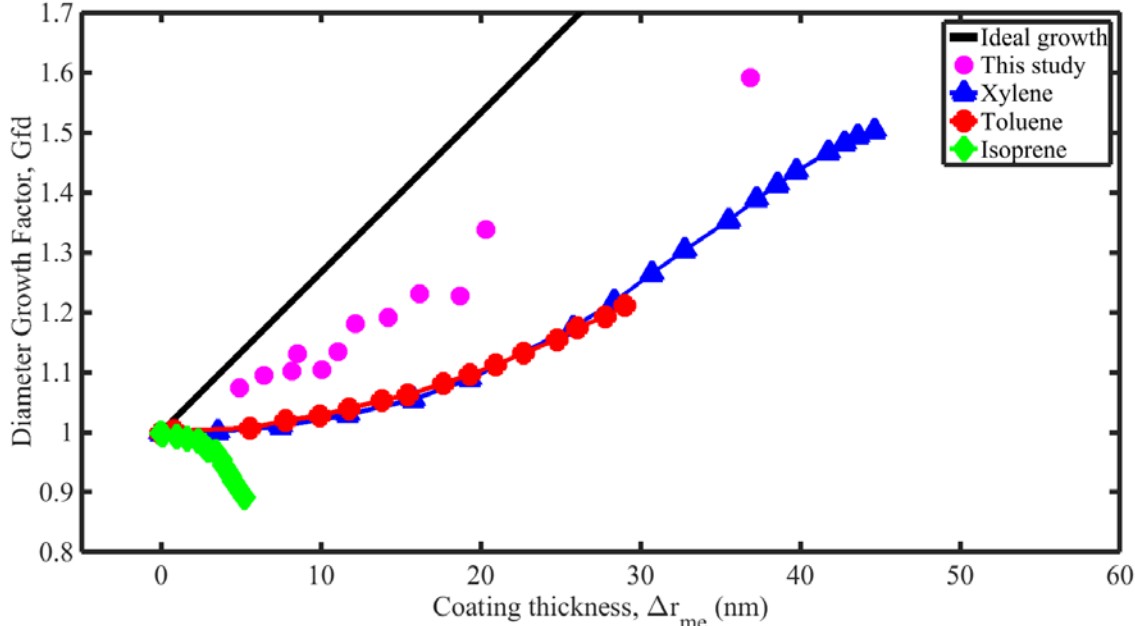

**Figure 5. Mobility diameter growth factor (*Gfd*) with initial fresh soot mobility diameter of 100 nm as a function of the mass equivalent coating thickness ($\Delta r_{me}$) reported in the literature and obtained in this study.**




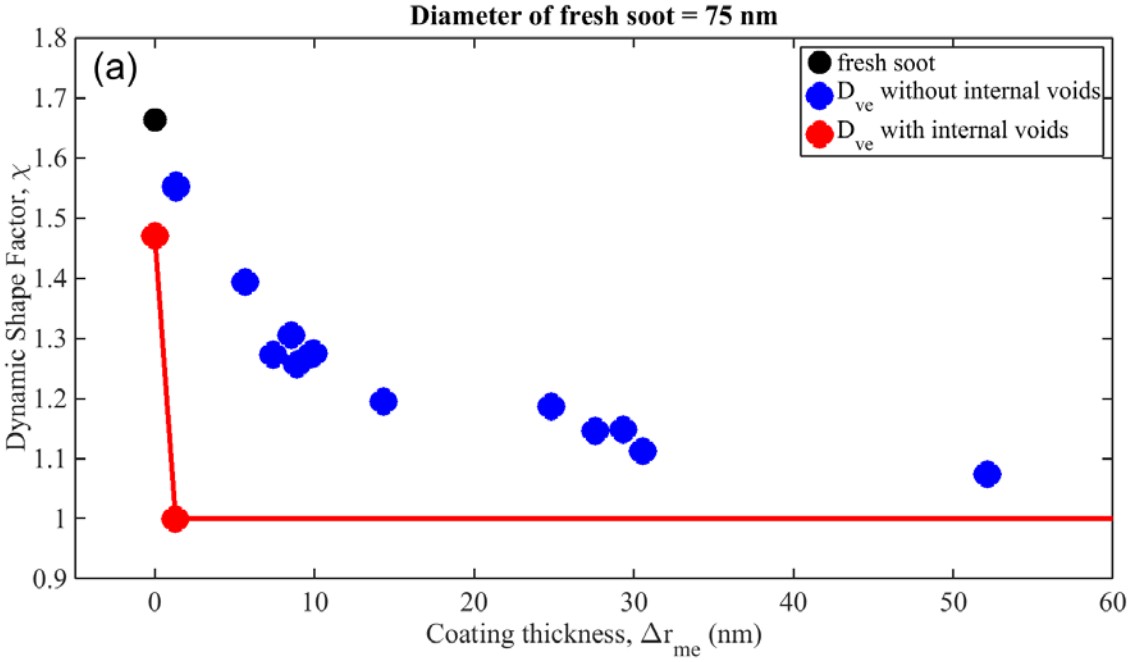

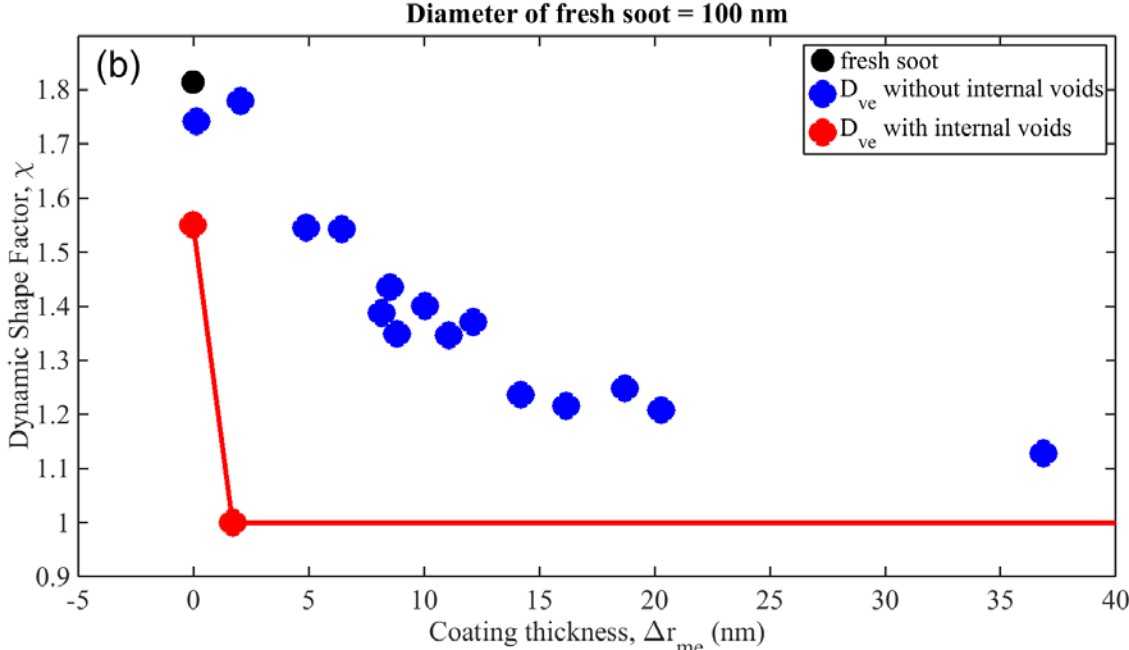





Figure 6. Dynamic shape factor ($\chi$) of fresh and processed soot particles as a function of the mass equivalent coating thickness ($\Delta r_{me}$).





**Figure 7. Effective density ($\rho_{eff}$) of fresh and processed soot particles as a function of the mass equivalent coating thickness ($\Delta r_{me}$).**

