# Peer review of "Morphological transformation of soot: investigation of microphysical processes during the condensation of sulfuric acid and limonene ozonolysis product vapors"

_Atmospheric Chemistry and Physics, 2017_

## Referee Comment (RC1) · Anonymous Referee #1 · 15 Jan 2018

This work presents the morphological transformation of soot during condensation of sulfuric acid and limonene SOA. Besides the present work proposes a framework (method) to quantify the parameters of this morphological transformation, i.e. amount of material used for filling voids or diameter growth and fractions of internal/open voids. Overall, this manuscript is well organized and written, the results are clearly presented, and the scientific novelty is significant for the society. However, the MS still needs minor revision and some technical modification. After that, I believe this paper can be published on ACP.

[Figure]

Comments: 1. Abstract: some quantitative result should be added in the abstract rather than general description. 2. The expression "framework" is strange. 3. Line 11: "soot. This work constitutes the first study that quantitatively tracks in-situ microphysical changes in soot morphology". I don't think it's true. 4. Page 1, Line 18 and in the whole manuscript. It is better to use "soot aggregate", not "soot agglomerate" to keep consistent with most of the literature. 5. Page 1, Line 32: Change "Growth" to "growth". 6. Page 2, Line 1-2: change ' ' to " " 7. Page 5, Line 27-30: this paragraph describes the steps of the experiments, however, the experiment of soot coated with only SOA is ignored, it should be stated clearly. 8. Page 7, Line15-16: Kuwata et al (2012) did not report this 1.26 number but provide a method to calculate the density, this sentence should be checked. 9. Page 7, Line 18: change the reference format to Saathoff et al., (2009) 10. Page 11, Line 6: should be sections 3.3 and 3.4 11. Table 3. I suggest that the authors should compare their work with other studies, e.g. Khalizov et al., EST 2013 etc. 12. Figure 1. The results of SP-AMS and CCN counter were not reported in this paper, should be removed from the figure. 13. Figure 6 (a-d): in the abstract, the author state that "In fact, most of the fresh soot particles considered in this study were largely spherical (dynamic shape factor: $\sim$1.1)", however, in this figure, the dynamic shape factors of fresh soot with internal voids are around 1.5-1.9, please check the data consistency. And the black dots in the figure should be changed to blue color as other points without internal voids, or in the legend change "fresh soot" to "fresh soot without internal voids" to make it more clear. 14. The figure captions are too brief. The author should explain more to make reader better understand the figures. i.e. Figure 2 needs to explain what do S, M, L mean etc.

Please also note the supplement to this comment:
https://www.atmos-chem-phys-discuss.net/acp-2017-769/acp-2017-769-RC1-supplement.pdf

---

## Referee Comment (RC2) · Anonymous Referee #2 · 20 Feb 2018

The authors present data on measurements on the mass equivalent and mobility size of fresh and coated soot particles, which is then interpreted using a framework that can explain the sequential transformations observed. The theoretical treatment is interesting and useful, and the conclusions are for the most part consistent with the data analysis. The subject is very suitable to ACP, and therefore I would suggest publication of the paper. There are some corrections and clarification that need to be made before publication, and I have also a few suggestions to improve readability.

Abstract (and conclusion): In the abstract, it is stated that the dynamic shape factor

of fresh soot was in most cases ca. 1.1; this is also stated in the conclusions. This is clearly wrong, as none of the values reported e.g. in Table 1 are even close to 1.1. Please correct.

Abstract (p2, l11): I'm not sure that it is correct to say that this is the first study to track microphysical changes in situ, as e.g. observations of changes in soot effective density have been made for a long time.

p 4, l34: "(ii) volume equivalent inclusive of internal voids": just to clarify; are the internal voids assumed to be part of the particle volume, but external voids are not? If yes, this could be clarified in the explanation of the framework to make following it easier.

p7, eq. 8; when calculating the mass equivalent density of a particle, which density is used? Does this translate also to the mass equivalent coating thickness? This could be useful to indicate, because a person using the framework will not know which density (effective, sulfuric acid, SOA, etc...) to use.

p9, l9; I did not fully understand what the difference between the nominal and actual mobility diameters are. They are selected with the same instrumentation and if nothing is done to the aerosol in between, they should be the same? Please clarify.

p9, l17: I'm a little bothered by the use of 'preferentially' in the paper. If I understand the text correctly, the open voids are filled first (shown by horizontal lines in figs (4), and then the particles start to grow. Are the internal voids filled at all? To my understanding, the internal voids are assumed to be left open (in the framework at least). This could be stated more clearly.

p11, l 27: "The black, red, and green colors in each pie chart represent the mass fraction of black carbon, sulfuric acid, and organics, respectively". How were these mass fractions obtained? Also, this information should be in the caption.

page 12, l25: Move the part starting with 'We assume...' and ending with 'in this work'

to the start of the explanation of the framework, as it will clarify the explanation better than here.

page 12, line 33: The phrase step-wise filling is often mentioned. I understood that there are basically two steps: void filling, and subsequent growth. Are there more? The collapse of the structure is also mentioned at some point, but this is not shown in Fig. 2. I would suggest that the actual steps are explicitly marked and named in at least one of Figs (4), preferentially all. Also, they should be explained in more detail in the captions.

Page 15, line 20-22: …" This may introduce some shift in the mapping of the mass and mobility size, leading to overestimation of Δrme." I don't really understand how the continuous growth causes a shift in the measured mobility or mass; please clarify this. Also, is there a reason why the soot differs so much in the internal/open void properties between the present and literature studies in Table 3?

Conclusions, p. 17, l21: '...this is the first study that considers the effect of of coatings with two chemical components'. Is there any conclusions drawn on the effect of the different components, and which properties cause these differences? I could not find these, and as this is not the main purpose of the paper, maybe this sentence could be changed.
* * *

---

## Author Response (AR1)

**Answers to the review of anonymous Referee #1**

We thank Referee #1 for reviewing our manuscript and giving useful suggestions. Below, comments from the referee are given in blue while our answers are given in black. In addition, the new text is marked blue in the revised version of the manuscript.

Review of Pei et al.

This work presents the morphological transformation of soot during condensation of sulfuric acid and limonene SOA. Besides the present work proposes a framework (method) to quantify the parameters of this morphological transformation, i.e. amount of material used for filling voids or diameter growth and fractions of internal/open voids. Overall, this manuscript is well organized and written, the results are clearly presented, and the scientific novelty is significant for the society. However, the MS still needs minor revision and some technical modification. After that, I believe this paper can be published on ACP.

Comments:

1. Abstract: some quantitative result should be added in the abstract rather than general description.

Response: We agree with the referee that some quantitative results should be added.

**Action:** One sentence "For the largest coating experiments, 6%, 10%, 24% and 58% of condensed material went to filling process, while 94%, 90%, 76% and 42% of condensed material went to growth process for 75 nm, 100 nm, 150 nm and 200 nm soot particles, respectively." has been added. (See page 1, line 28-31 in the revised manuscript).

2. The expression "framework" is strange.

Response: This is new word in context to morphological transformation of soot aggregate. The methodology of quantifying the filling of voids and growth of mobility diameter with respect to ideal line is referred as "framework". The framework here is method to quantify. "Framework" seems the most appropriate word in this context.

**Action:** No change.

3. Line 11: "soot. This work constitutes the first study that quantitatively tracks in-situ microphysical changes in soot morphology". I don't think it's true.

Response: This is a work that had quantified the in-situ morphological transformation of soot aggregate i.e. filling of the voids and growth of particle, nevertheless we agree with the referee that the morphological transformation of soot aggregate has been studied in the previous studies.

**Action:** We have removed word "first" and modified the sentences wherever appropriate. (See page 2, line 6 and page 18, line 26 in the revised manuscript).

4. Page 1, Line 18 and in the whole manuscript. It is better to use "soot aggregate", not "soot agglomerate" to keep consistent with most of the literature.

Response: We agree with the referee.

**Action:** We have changed all "soot agglomerate" to "soot aggregate" in the whole manuscript.

5. Page 1, Line 32: Change "Growth" to "growth".

Response: We agree with the referee.

**Action:** Done. (See page 1, line 24 in the revised manuscript).

6. Page 2, Line 1-2: change ' ' to " "

Response: We agree with the referee.

**Action:** Done. (See page 1, line 26 in the revised manuscript).

7. Page 5, Line 27-30: this paragraph describes the steps of the experiments, however, the experiment of soot coated with only SOA is ignored, it should be stated clearly.

Response: We agree with the referee the description of experiment of SOA coated soot should be added.

**Action:** The whole sentence is changed to "The experimental matrix included four major procedural steps: 1) characterization of the pure soot core; 2) modification of the soot core surface with sulfuric acid and characterization of the modulated soot; 3) modification of soot core surface with SOA and characterization of the modulated soot; 4) modulation of the sulfuric acid modified soot with limonene ozonolysis products and subsequent characterization." (See page 5, line 24-28 in the revised manuscript).

8. Page 7, Line15-16: Kuwata et al (2012) did not report this 1.26 number but provide a method to calculate the density, this sentence should be checked.

Response: We agree with the referee that the statement is not clear.

**Action:** the sentence "This value is similar to the organic aerosol density $(1.26\pm0.04$ g cm$^{-3}$) estimated from O:C and H:C determined via SP-AMS measurements performed by Kuwata et al. (2012), and the value $(1.3\pm0.2$ g cm$^{-3}$) reported by Saathoff et al., (2009). " has been changed to "In this study,

aerosol mass spectrometer (AMS) was also used in parallel with DMA-APM system. The O:C and H:C determined via AMS measurements were used to estimate the organic aerosol density ($1.26\pm0.04$ g cm$^{-3}$) with the method given by Kuwata et al. (2012). The material density of limonene ozonolysis products (1.2 g cm$^{-3}$) used in this study is similar to the AMS results and the value ($1.3\pm0.2$ g cm$^{-3}$) reported by Saathoff et al., (2009)." (See page 7, line 14-18 in the revised manuscript).

9. Page 7, Line 18: change the reference format to Saathoff et al., (2009)

Response: We agree with the referee.

**Action:** Done. (See page 7, line 17-18 in the revised manuscript).

10. Page 11, Line 6: should be sections 3.3 and 3.4

Response: We agree with the referee.

**Action:** Done. (See page 11, line 15 in the revised manuscript).

11. Table 3. I suggest that the authors should compare their work with other studies, e.g. Khalizov et al., EST 2013 etc.

Response: We thank for the referee's suggestion. However, Khalizov et al. (2013) does not present their results of the diameter growth factor (*Gfd*) as function of coating thickness ($\Delta r_{me}$), (-$\Delta r_{me}$ *for them cannot be* estimated to compare with us) and the maximum coating thickness in this literature is only 6 nm. These two factors make **it difficult to** compare the results of two studies.

**Action:** No change.

12. Figure 1. The results of SP-AMS and CCN counter were not reported in this paper, should be removed from the figure.

Response: We agree with the referee that SP-AMS and CCN counter are not necessary in this paper.

**Action:** Done.

13. Figure 6 (a-d): in the abstract, the author state that "In fact, most of the fresh soot particles considered in this study were largely spherical (dynamic shape factor: ~1.1)", however, in this figure, the dynamic shape factors of fresh soot with internal voids are around 1.5-1.9, please check the data consistency. And the black dots in the figure should be changed to blue color as other points without internal voids, or in the legend change "fresh soot" to "fresh soot without internal voids" to make it more clear.

Response: We agree with the referee that the values are not consistent. We have checked the data carefully and we found that in the figures the values are wrong due to a mistake in calculation. The values should be in the range of 1.03-1.08.

**Action:** Figure 6 has been updated with correct values in the revised manuscript. In addition, the values in Table 1 have also been corrected. The legend for the black dots in Figure 6 has also been changed to "fresh soot without internal voids" according to the referee's suggestion.

**Answers to the review of anonymous Referee #2**

We thank Referee #2 for reviewing our manuscript and giving useful suggestions. Below, comments from the referee are given in blue while our answers are given in black. In addition, the new text is marked blue in the revised version of the manuscript.

The authors present data on measurements on the mass equivalent and mobility size of fresh and coated soot particles, which is then interpreted using a framework that can explain the sequential transformations observed. The theoretical treatment is interesting and useful, and the conclusions are for the most part consistent with the data analysis. The subject is very suitable to ACP, and therefore I would suggest publication of the paper. There are some corrections and clarification that need to be made before publication, and I have also a few suggestions to improve readability.

1. Abstract (and conclusion): In the abstract, it is stated that the dynamic shape factor of fresh soot was in most cases ca. 1.1; this is also stated in the conclusions. This is clearly wrong, as none of the values reported e.g. in Table 1 are even close to 1.1. Please correct.

Response: We agree with the referee, the dynamic shape factor in the abstract and conclusions are not same as values in Table 1 and Figure 6. However, after checked the data and calculation carefully, we found that the values in Table 1 and Figure 6 are wrong due to a mistake in calculation. The statement and conclusion on the morphology of fresh soot particles in this study remain valid.

**Action:** The sentence in the abstract has been changed to "In fact, the dynamic shape factor adjusted for internal voids was close to 1 for the fresh soot particles considered in this study, indicating the particles were largely spherical." (See page 2, line 2–4 in the revised manuscript). The sentence in the conclusion has been changed to "In fact, the dynamic shape factor adjusted for internal voids ($\chi_i$) was close to 1 for the fresh soot particles considered in this study, indicating the particles were largely spherical." (See page 18, line 20–21

in the revised manuscript). The values in Table 1 are corrected and Figure 6 is also updated accordingly.

2. Abstract (p2, l11): I'm not sure that it is correct to say that this is the first study to track microphysical changes in situ, as e.g. observations of changes in soot effective density have been made for a long time.

Response: This is a work that had quantified the in-situ morphological transformation of soot aggregate i.e. filling of the voids and growth of particle, nevertheless we agree with the referee that the morphological transformation of soot aggregate has been studied in the previous studies.

**Action:** We have removed word "first" and modified the sentences wherever appropriate. (See page 2, line 6 and page 18, line 26 in the revised manuscript).

3. p 4, l34: "(ii) volume equivalent inclusive of internal voids": just to clarify; are the internal voids assumed to be part of the particle volume, but external voids are not? If yes, this could be clarified in the explanation of the framework to make following it easier.

Response: We agree with the referee. The internal voids are assumed to be part of the particle volume.

**Action:** The sentence "(ii) determining the volume equivalent diameter inclusive of internal voids." has been changed to "(ii) determining the volume equivalent diameter inclusive of unfilled voids." (See page 4, line 28 in the revised manuscript).

4. p7, eq. 8; when calculating the mass equivalent density of a particle, which density is used? Does this translate also to the mass equivalent coating thickness? This could be useful to indicate, because a person using the

framework will not know which density (effective, sulfuric acid, SOA, etc...) to use.

Response: We agree with the referee, the explanation of the density should be clear. The material density is used to calculate the mass equivalent coating thickness.

**Action:** In response to address the issue raised by the referee, one sentence "For fresh soot $\rho_m$ is the material density of the soot, whereas for coated particle $\rho_m$ is the average material density over all the components of the particle, which can be calculated from Eq. (4)–(7)." has been added after Eq. (8). (See page 8, line 1-2 in the revised manuscript).

5. p9, l9; I did not fully understand what the difference between the nominal and actual mobility diameters are. They are selected with the same instrumentation and if nothing is done to the aerosol in between, they should be the same? Please clarify.

Response: The nominal mobility diameter is the setting value of the first DMA, whereas the actually mobility diameter is the value measured with the second DMA. These are the digital values by two identical sets of instruments within the instrumental noise/error. So in reality both mean the same.

6. p9, l17: I'm a little bothered by the use of 'preferentially' in the paper. If I understand the text correctly, the open voids are filled first (shown by horizontal lines in figs (4), and then the particles start to grow. Are the internal voids filled at all? To my understanding, the internal voids are assumed to be left open (in the framework at least). This could be stated more clearly.

Response: We agree with the referee that open voids are filled first, but the filling of internal voids and particle growth can happen sequentially.

**Action:** The word "preferentially" appears 3 times in the original version of the manuscript, and the word "preferential" appears once in the original version of the manuscript.

On page 9, line 17 in the original version, the sentence "(i) in the case of SOA, the open voids in the condensed material are preferentially filled prior to the onset of growth" has been changed to "(i) in the case of SOA, the open voids are filled prior to the onset of growth". (See page 9, line 21 in the revised manuscript).

On page 12, line 29 in the original version, the word "preferentially" has been removed. (See page 13, line 6 in the revised manuscript).

On page 12, line 30 in the original version, the word "preferential" has been removed in the revised version. (See page 13, line 7 in the revised manuscript).

On page 14, line 15 in the original version, the sentence "…, the filling of internal voids occurs preferentially to growth by high surface tension species, …" has been changed to "…, the filling of internal voids occurs prior to growth by high surface tension species, …". (See page 14, line 27 in the revised manuscript).

7. p11, l 27: "The black, red, and green colors in each pie chart represent the mass fraction of black carbon, sulfuric acid, and organics, respectively". How were these mass fractions obtained? Also, this information should be in the caption.

Response: The mass fractions of black carbon, sulfuric acid and organics are calculated from APM measurements. The method to calculate these mass fractions is described in the section 2.2 Data analysis part, and given by Eq. (4)–(6).

**Action:** In the caption of Figure 4, one sentence "The black, red, and green colors in each pie chart represent the mass fraction of black carbon, sulfuric acid, and organics calculated from Eq. (4)–(6), respectively" has been added.

8. page 12, l25: Move the part starting with 'We assume...' and ending with 'in this work' to the start of the explanation of the framework, as it will clarify the explanation better than here.

Response: We think that it reads quite well here as well.

**Action:** No change.

9. page 12, line 33: The phrase step-wise filling is often mentioned. I understood that there are basically two steps: void filling, and subsequent growth. Are there more? The collapse of the structure is also mentioned at some point, but this is not shown in Fig. 2. I would suggest that the actual steps are explicitly marked and named in at least one of Figs (4), preferentially all. Also, they should be explained in more detail in the captions.

Response: there are only two basic steps: void filling and particle growth illustrated in the framework. However, in reality, void filling may lead to collapse of the structure (decrease in mobility diameter). We agree with the referee that actual steps should be marked and named in Figure 4.

**Action:** To give an example, actual steps including voiding filling, particle growth and collapse are marked by arrows and named in Figure (4d). The explanation of the purple lines "Purple lines parallel to the ideal sphere growth line (dashed black) represent growth of the particle diameter; purple lines parallel to the $x$-axis represent filling of voids; purple lines with negative slope indicates a combination of void filling and collapse of the soot particle." has been added in the caption.

10. Page 15, line 20-22: . . ." This may introduce some shift in the mapping of the mass and mobility size, leading to overestimation of $\Delta r_{me}$." I don't really understand how the continuous growth causes a shift in the measured mobility

or mass; please clarify this. Also, is there a reason why the soot differs so much in the internal/open void properties between the present and literature studies in Table 3?

Response: In smog chamber experiments, SOA composition is changing continuously because it is constantly evolving from reactions and aging of SOA. In fact, SOA condensed on soot is not the same in any two measurements. So we think that the continuous growth of soot in smog chamber may be different for two consecutive points since it took about 20-30 min to acquire two measurement points. In other words, during the measurement the particles in the environmental chamber were evolving and there are several factors that can cause the differences between this study and literature studies:

1) The coating devices and time scales were different: in this study the coating device was laminar flow reactor and the residence time was 4.8 min while in the literature studies the coating device was a collapsible environmental chamber and the time for each their experiment was several hours.

2) The soot was different: in this study the soot was generated from a premixed-diffusion flame and denuded with a thermos-denuder at 400 °C, while in the literature studies the soot was generated from a Santoro-type laminar diffusion burner without thermo-denuder.

3) The condensed materials were different: in this study the coating materials were two types: sulfuric acid and limonene ozonolysis SOA, while in the literature studies the condensed materials were only SOA: toluene OH oxidation products in Qiu et al. (2012), and m−xylene OH oxidation products in Guo et al. (2016).

11. Conclusions, p. 17, l21: '...this is the first study that considers the effect of coatings with two chemical components'. Is there any conclusions drawn on the effect of the different components, and which properties cause these differences? I could not find these, and as this is not the main purpose of the paper, maybe this sentence could be changed.

Response: We agree with the referee that the statement on the effect of coatings with two chemical components is not clear.

**Action:** The sentence "To the best of our knowledge, this is the first study that considers the effect of coatings with two chemical components (i.e., sulfuric acid and SOA) on soot morphology." has been removed. (See page 18, line 9 in the revised manuscript).

**Changes according to co-authors' suggestions**

We also change some text in the manuscript according to our co-authors' suggestions, and they are marked in blue in the revised manuscript with makeup. The page numbers and line numbers are according to the revised version with makeup.

Page 1, line 4: In "Joakim H. Pagels", "H." is removed.

Page 1, line 19: "discovered" is changed to "hypothesize".

Page 1, line 31-33: "We also used the framework to estimate the fraction of internal voids and open voids. This information was then used to derive the volume equivalent diameter of the soot aggregate containing internal voids and to calculate the *in-situ* dynamic shape factor" is changed to "We also used the framework to estimate the fraction of internal voids and open voids. This information was then used to estimate the volume equivalent diameter of the soot aggregate containing internal voids and to calculate the dynamic shape factor, accounting for internal voids."

Page 2, line 2: "*in-situ*" is removed.

Page 2, line 2-4: "In fact, most of the fresh soot particles considered in this study were largely spherical (dynamic shape factor: 1.03–1.08)." is changed to "In fact, the dynamic shape factor adjusted for internal voids was close to 1 for

the fresh soot particles considered in this study, indicating the particles were largely spherical."

Page 2, line 7-8: "This work constitutes the first study that quantitatively track *in-situ* microphysical changes in soot morphology,…" is change to "In this work we quantitatively tracked *in-situ* microphysical changes in soot morphology,…"

Page 2, line 27: "…, aged hygroscopic soot is more efficiently deposited in the lungs" is changed to ".., aged hygroscopic soot has an altered deposition in the lungs".

Page 3, line 6: "…, and proportionality constant, respectively." is change to "…, and a proportionality constant, respectively."

Page 8, line 4-5: "For fresh soot $\rho_m$ is the material density of the soot, whereas for coated particle $\rho_m$ is the average material density over all the components of the particle, which can be calculated from Eq. (4)–(7)." is changed to "For fresh soot $\rho_m$ is the inherent material density of the soot, whereas for coated particle $\rho_m$ is the average inherent material density over all the components of the particle, which can be calculated from Eq. (4)–(7)."

Page 8, line 5-7: Sentence "It should be pointed out that this definition of the mass equivalent diameter is identical to the definition of volume equivalent diameter used by McMurry et al. (2002) and Park et al. (2003)." is added.

Page 8, line 13-18: The text is changed to "

The dynamic shape factor $\chi$ can be calculated from the measured mobility diameter $D_p$ and the mass equivalent diameter $D_{me}$ (Baron and Willeke, 2001):

$$\chi = \frac{D_p C_{me}}{D_{me} C_p},\tag{10}$$

where $C_{me}$ and $C_p$ are the Cunningham slip correction factors for particles with diameters $D_{me}$ and $D_p$, respectively. The dynamic shape factor is derived directly from Stokes' law. It is the ratio of the drag force exerted on the irregular particle divided by the drag force exerted on its mass equivalent sphere, when travelling at the same speed."

Page 8, line 18-20: "In situ experimental determination of $D_{ve}$ is not currently possible and, hence, soot aggregates free of internal voids are typically assumed, leading to $D_{ve} = D_{me}$." is removed.

Page 8, line 22-23: "The void space fraction ($F_{vs}$), i.e., volume of voids/total volume of particles derived from the mobility diameter, is calculated from the $D_{me}$ and $D_p$ of fresh and coated soot:" is changed to "The void space fraction ($F_{vs}$), i.e., volume of voids/total volume of particles derived from the mobility

diameter, is estimated from the $D_{me}$ and $D_p$ of fresh and coated soot (Baron and Willeke, 2001; Zhang et al., 2016):"

Page 9, line 26: "in the condensed material" is removed.

Page 10, line 4-6: the text "

Then $D_{ve,i}$ is used to calculate the dynamic shape factor with internal voids ($\chi_i$):

$$\chi_i = \frac{D_p C_{ve,i}}{D_{ve,i} C_p} , \qquad (20)$$

where $C_{ve,i}$ is corresponding Cunningham slip correction factors for $D_{ve,i}$." is added.

Page 10, line 23-24: "However, compared with previous studies, our quantification considers internal voids (which in fact results in significantly smaller $\chi$) and therefore yields more accurate $\chi$ (see section 3.4)." is removed.

Page 10, line 28 – page 11, line 1: "This value is consistent with that reported for fresh soot particles from a diffusion propane burner" is changed to "This value is consistent with aggregates reported for fresh soot particles from a diffusion propane burner"

Page 11, line 22: "fractality " is changed to "porosity".

Page 11, line 24-25: "In previous studies, $\chi$ was calculated based on the assumption that internal voids were absent from the soot aggregate. This assumption yields $D_{me} = D_{ve}$. In this work, $D_{ve}$ increases whereas $\chi$ decreases with the occurrence of internal voids (see sections 3.3 and 3.4 for the experimentally determined open-void fractions and relevant discussions)." is changed to "In previous studies, $\chi$ was calculated based on  $D_{me} = D_{ve}$. In this work, $D_{ve}$ increases whereas $\chi_i$ decreases when we attempt to adjust for internal voids (see sections 3.3 and 3.4 for the experimentally determined open-void fractions and relevant discussions).

Page 11, line 32 – page 12, line 1: "S, M, L" is changed to "small, medium, large" and they are all changed in the whole manuscript.

Page 15, line 18: "200 nm soot particle with a mass…" is changed to "200 nm soot particles with a mass…"

Page 15, line 33-34: "Detailed estimates associated with each step of filling and shrinkage of the 200 nm soot are provided in Table S2 of the supplement." is changed to "Detailed estimation associated with each step of filling and shrinkage of the 200 nm soot is provided in Table S2 of the supplement."

Page 16, line 8: "…leading to overestimation of $\Delta r_{me}$." is changed to "…leading to either underestimation or overestimation of $\Delta r_{me}$."

Page 16, line 22-23: "However, as previously stated, our experimental results show that internal voids dominate the total void space in all four cases." is changed to "However, as previously stated, our experimental results when interpreted with the new framework suggest that internal voids dominate the total void space in all four cases."

Page 16, line 24-26: "Therefore, the $\chi$ values obtained based on the *no internal voids* assumption differ significantly (see Fig. 6) from the experimentally determined values obtained in this study." is changed to "Therefore, the $\chi$ values obtained based on the *no internal voids* assumption ($\chi_n$) differ significantly (see Fig. 6) from the values where we attempted to adjust for internal voids ($\chi_i$)."

Page 16, line 28-31: "This assumption neglects the *in-situ* morphology of the soot aggregate, and stipulates (based on the notion of a void-less sphere) that the equivalent volume is equal to the sum of the all primary spherules. These findings highlight the serious shortcomings of the assumption that $D_{ve} = D_{me}$ and the implications for atmospheric surface processes that are considered critical for modeling-based studies." is changed to "This assumption stipulates (based on the notion of a void-less sphere) that the equivalent volume is equal to the sum of all the primary spherules. These findings have implications for atmospheric surface processes that are considered critical for modeling-based studies."

Page 16, line 31 – page 17, line 3": "As shown in Fig. 6(a)–(d), a significant amount of material is condensed on the soot particles, but a perfect sphere remains elusive. For e.g., the thickness of the coating on the 75 nm particle is at least two times larger than the initial mobility diameter, but $\chi$ still deviates from unity (Exp. 15). This results from the fact that $\chi$ is estimated (Eq. 10; $D_{ve} = D_{me}$) based on the assumption that the internal voids measured in our experiment are all open voids. " is changed to "As shown in Fig. 6(a)–(d), a significant amount of material is condensed on the soot particles, but a perfect sphere remains elusive. For e.g., the thickness of the coating on the 75 nm particle is at least two times larger than the initial mobility diameter, but $\chi_n$ still deviates from unity (Exp. 15). This results from the fact that $\chi_n$ is calculated (Eq. 10) based on the assumption that the voids measured in our experiment are all open voids. "

Page 17, line 5-9: "The $\chi$ values determined in previous studies (see Table S4) based on the *no internal voids* assumption appear to be consistent with each other for the wrong reason. In reality, and as confirmed in this study, the occurrence of internal voids in a soot aggregate is unavoidable. Therefore, we suggest that the framework introduced in this work should be developed using an experimental setup, i.e., a flow tube integrated with a DMA-APM. This setup will yield *in-situ $D_{ve}$* associated with the morphological characteristics of soot and the transformation of these characteristics upon the condensation of material." is removed.

Page 17, line 9-18: "Our framework highlights that the high values of the dynamic shape factor found at high coating thicknesses can not only be caused by truly non-spherical particles (for example a few chains of the soot core sticking out from a spherical droplet) but in addition internal porosity in the soot core that blocks full penetration of the condensed species. Thus, internal porosity causes uncertainty when judging the particle shape with conventional approaches. The main uncertainty in the new frame-work comes from the determination of $F_{vs}$ and $F_i$. For example, when assigning a volume fraction of voids based on mobility measurements and when dividing $F_{vs}$ into internal ($F_i$) and open void space based on the growth curves. In addition, when interpreting the data using our framework one finds that the low values of $Dfm$ as well as size-dependent effective density do not always indicate soot aggregates have an open-structure, but instead compact soot cores with increasing internal porosity can also have these features and match the mass-mobility data." is added.

Page 17, line 23: "lines" is changed to "curves".

Page 17, line 24-25: "The method of calculating the ideal growth curve is described in detail in the supplement." is added. The detail information is added in the supplement accordingly.

Page 18, line 26: "on" is changed to "due to".

Page 19, line 2-4: "The dynamic shape is also calculated from the parameters derived by the framework. The dynamic shape factor estimated from traditional assumptions and methods differs significantly from the value determined in this study." is changed to "The dynamic shape factor calculations where we attempted to account for internal voids ($\chi_i$) is also calculated from the parameters derived by the framework. The dynamic shape factor estimated from traditional assumptions and methods ($\chi_n$) differs significantly from the value determined in this study."

Page 19, line 4-6: "In fact, most of the fresh soot particles considered in this study are largely spherical, with a dynamic shape factor of 1.03–1.08." is changed to "In fact, the dynamic shape factor adjusted for internal voids ($\chi_i$) was close to 1 for the fresh soot particles considered in this study, indicating the particles were largely spherical"

Page 19, line 11: "represents first study that" is removed.

Several new references are added in the references list.

[revised manuscript text omitted]